# Provable Benefits of RLVR over SFT for Reasoning Models: Learning to Backtrack Efficiently

**Stanley Wei** [1]   **Juno Kim** [2]

## Abstract

Recent advances in large language models (LLMs) have demonstrated that reinforcement fine-tuning of pretrained base models can lead to significant gains in reasoning performance at inference time. In this work, we theoretically analyze why reinforcement fine-tuning induces better reasoning ability than purely supervised fine-tuning (SFT) methods. We model chain-of-thought (CoT) reasoning as a pathfinding problem on graphs and compare the popular method of reinforcement learning with verifiable rewards (RLVR) against traditional SFT. We prove that SFT, when trained on golden shortest paths without negative examples, fails to learn how to efficiently backtrack. In contrast, an RLVR-trained model can learn how to efficiently backtrack from dead ends using only outcome reward. This leads to an exponential separation in inference-time compute between the two methods, and demonstrates that RLVR leads the model to learn the location of difficult decisions in a reasoning chain, ultimately allowing for better allocation of inference-time compute. Finally, we show that the reasoning traces of an RLVR model can be distilled to train a base model to backtrack efficiently as well.

## 1. Introduction

Modern large language models (LLMs) are trained to achieve strong reasoning capabilities on a wide range of tasks such as mathematical problem-solving and code generation. In the context of LLMs, *reasoning* refers to the generation of long chain-of-thought (CoT) which mimic the step-by-step nature of human reasoning to solve complex logical problems (Wei et al., 2022; Lightman et al., 2023).

Reasoning models treat such tasks as a sequential multi-step decision process and deploy strategies utilizing additional test-time compute budget, such as sampling, tree search, aggregation, and backtracking. This approach has proved to yield efficient and scalable gains over initial pretraining (Snell et al., 2024; Muennighoff et al., 2025), and has been adopted with great success in various frontier and open-source models (OpenAI, 2024; Shao et al., 2024; Guo et al., 2025).

In order to learn or strengthen these desired inference-time reasoning behaviors, LLMs must undergo stages of post-training (Kumar et al., 2025; Xu et al., 2025). The post-training procedure typically follows one of two main strategies: supervised fine-tuning (SFT) on expert demonstrations, and reinforcement learning (RL) on feedback from a reward model or task verifier. SFT uses an off-policy dataset of demonstrations created or annotated by an expert (typically a human or a more powerful model) and trains the base model to imitate these responses. While SFT has been the de facto method for post-training, it is prone to memorization or overfitting (Chu et al., 2025a) and can induce uninformative pseudo-reasoning paths (Chen et al., 2025). Moreover, curating high-quality expert demonstrations can be expensive and time-consuming, depending on the nature of the task.

In contrast, methods such as reinforcement learning with verifiable rewards (RLVR) (Wen et al., 2025; DeepSeek-AI et al., 2025a) and reinforcement learning from human feedback (RLHF) (Christiano et al., 2023; Ouyang et al., 2022) learn from reward models or verifiers via on-policy exploration. The RL approach has been argued to generalize more effectively and have the potential to unlock entirely new reasoning capabilities (Chu et al., 2025a; Wang et al., 2025; Zhu et al., 2025). Nevertheless, we still lack a principled understanding of the differences between the two methods, or a theoretical framework under which to compare various post-training algorithms.

In this paper, we focus on *backtracking ability* as a way to distinguish the effectiveness of post-training methods. Backtracking is a key element of human problem-solving: when a line of approach is revealed to be incorrect or unhelpful, one returns to an earlier decision point and explores an

---

[1] Princeton University [2] University of California, Berkeley. Correspondence to: Stanley Wei <stanley.wei@princeton.edu>.

*Proceedings of the 43rd International Conference on Machine Learning*, Seoul, South Korea. PMLR 306, 2026. Copyright 2026 by the author(s).

alternative branch. This greatly reduces complexity of the effective search space. Empirically, backtracking has been shown to be greatly beneficial to LLM reasoning ability, either implicitly in the chain of thought (Cai et al., 2025), or explicitly with backtrack tokens (Yang et al., 2025) or rolling back sequence generation (Singh et al., 2025). Hence we are motivated to ask:

*Which post-training method teaches the base model to efficiently backtrack at inference time?*

**Our contributions.** We approach this question by modeling CoT reasoning as pathfinding on graphs, a sandbox commonly studied in the literature (Sanford et al., 2024; Bachmann & Nagarajan, 2025; Kim et al., 2025). While existing works have studied the benefits of backtracking from an information or sampling perspective (Shalev-Shwartz & Shashua, 2025; Rohatgi et al., 2025), we provide a novel *dynamical* characterization of running SFT versus RLVR. We design a multigraph similar to the path-star graph (Bachmann & Nagarajan, 2025) with $W$ branches of depth $K$, which the pretrained world model – a linear-softmax bigram over edges, or a trigram over nodes – must learn to navigate by backtracking from failed branches. Our results are summarized as follows.

- We prove that SFT trained only on golden shortest paths does not learn any backtracking strategy, while the RLVR-trained model learns to backtrack consistently using only outcome reward with a length penalty in finite time.

- We show an exponential test-time compute separation: after convergence, the RLVR model can reach any target in $\Theta(WK)$ expected time, while the SFT model requires $\Theta(WL^K)$ time.

- We further show that distilling RLVR-generated reasoning traces to a base model via supervised learning transfers efficient backtracking, recovering $\Theta(WK)$ inference-time compute.

Intuitively, SFT on expert solutions trains the base model to continue along a gold path with no exposure to dead ends. Hence, SFT need not learn an efficient retreat strategy. In contrast, on-policy RLVR necessarily generates and trains on the model's own unsuccessful partial rollouts. This exploration produces gradient signal not only about good forward actions, but also backtracking actions when the model has committed to a poor branch. We formalize this difference via a dynamical analysis of gradient flow (for SFT) and sign policy-gradient flow (for RLVR). Our results provide theoretical support for the importance of backtracking data for reasoning in both RL and distillation.

All proofs are deferred to the appendix.

## 2. Related Work

**Backtracking in LLM reasoning.** Backtracking has empirically been used to quantify and improve reliability and efficiency in LLM reasoning. Inference-time search frameworks such as Tree-of-Thoughts explicitly explore and prune a branching space of intermediate thoughts, leading to depth-first or best-first search with backtracking (Yao et al., 2023; Long, 2023), generalized by Graph-of-Thoughts methods (Besta et al., 2024). Qin et al. (2025) study when sequential search with backtracking benefits over parallel sampling. In particular, they empirically show that models with backtracking capabilities benefit greatly from RL finetuning on reasoning tasks, which agree with our theoretical results. Moreover, Singh et al. (2025) propose preemptive backtracking guided by in-context value verification to identify and focus resampling from suspected failure points. Yang et al. (2025) study encouraging models to decide when to revert during reasoning via introducing an explicit backtrack token, aiming to internalize structured backtracking.

From a theoretical perspective, Shalev-Shwartz & Shashua (2025) show the necessity of search and backtracking for certain graph search tasks and parity problems, and describe a learning method which builds a search tree with explicit backtracking. Rohatgi et al. (2025) propose a test-time sampling and backtracking algorithm which uses process rewards, which mixes in time quadratic in the depth of the search tree, allowing for exact sampling from a target distribution on the leaves. However, these works do not study how to learn such behavior via post-training, which is the focus of our dynamical analysis. Kim et al. (2025) propose a cluster graph model of reasoning and analyze CoT pathfinding as a metastable Markov process. They also provide convergence guarantees for a simple RL method (proximal policy optimization); however, they do not study RLVR or SFT, and do not consider backtracking in CoT.

**RL versus SFT.** Recent empirical studies have highlighted significant qualitative differences between models post-trained via SFT and those with RL. Chu et al. (2025b) provide a comparative analysis on an arithmetic-based reasoning task that suggests SFT tends to memorize the distribution of training demonstrations, whereas RL facilitates better generalization to out-of-distribution (OOD) tasks. This distinction is further supported by Shenfeld et al. (2025), in which the authors argue that on-policy RL implicitly regularizes the model towards KL-minimal solutions (with respect to the base model), allowing the model to find simpler solutions that are robust to catastrophic forgetting. Park et al. (2025) also report that SFT underperforms RL on the synthetic Countdown task; moreover, RL-only post-training can induce fundamentally improved OOD generalization. In addition, Chen et al. (2025) show that SFT can elicit pseudo-reasoning paths in large vision-language models by

only superficially imitating expert models. Such paths often contain uninformative or incorrect reasoning steps, and even hurt subsequent RL training stages. They also propose an RL-based approach which leads to more adaptive reasoning behavior.

**Reasoning as graph search.** Pathfinding in graphs has been widely used as a sandbox for understanding LLM reasoning capabilities. Abbe et al. (2024) propose the notion of globality degree to capture transformer learning ability, and show that regular transformers cannot efficiently solve cycle tasks. Sanford et al. (2024) study the ability of transformer networks to solve various graph algorithms in terms of their expressivity such as network width and depth. From an empirical perspective, synthetic graph pathfinding tasks have been used to study compositional reasoning ability (Khona et al., 2024) and understand internal prediction mechanisms (Cohen et al., 2025). In particular, the path-star graph (which our construction generalizes) has been shown to be difficult to solve for LLMs, and has been used to study the limitations of reasoners trained via next-token prediction (Bachmann & Nagarajan, 2025; Frydenlund, 2024).

## 3. Problem Formulation

### 3.1. CoT as a pathfinding task

We model our reasoning task as pathfinding on a toy graph that represents the world model (Sanford et al., 2024; Bachmann & Nagarajan, 2025; Kim et al., 2025). Specifically, we consider a multigraph consisting of the following components; see Figure 1.

- Source node $s_0$, fork node $f$;
- Diamond $\Diamond(u, v)$, which is a subgraph of $L$ undirected multiedges between nodes $u, v$;
- Leaf nodes $t_i$.

The topology of the graph is as follows: the source node $s_0$ is connected to a fork node $f$ through a directed edge to $f$ (this will be the only directed edge in the graph). $f$ now branches out in $W$ directions. In each branch $i = 1, \cdots, W$, we have a sequence of $K$ *diamonds*:

$$\Diamond(u_{i,1,l}, u_{i,1,r}), \Diamond(u_{i,2,l}, u_{i,2,r}), \ldots, \Diamond(u_{i,K,l}, u_{i,K,r})$$

where consecutive left and right diamonds are connected. When multiedges occur, we will denote the edges by $u \overset{j}{\leftrightarrow} v$ for $1 \leq j \leq L$ for the multiedges in $\Diamond(u, v)$. Finally, we connect node $u_{i,K,r}$ to a leaf node $t_i$. For ease of notation, we will denote $u_{i,0,r} := f$ and $u_{i,K+1,l} := t_i$ for all $i$.

**Reasoning task.** In our setup, the reasoning model is prompted with a desired target node $u$, and the goal is to output a valid path from $s$ to $u$. We are interested in understanding how post-training a language model with the standard paradigm of supervised fine-tuning (SFT) versus reinforcement learning with verifiable rewards (RLVR) influences the test-time behavior of the model, when prompted with a new reasoning task.

### 3.2. Pretraining

We consider a bigram model, in which each state consists of an edge and traversal direction (*not* the underlying vertices); that is, for an edge $e := u \leftrightarrow v$ in the graph, $u \to v$ and $v \to u$ are valid states. Furthermore, valid state transitions from an edge $u \to v$ would be to states in the set $N_v := \{v \to w : v \leftrightarrow w \in E\}$ (in particular, note that multiedges are distinct). For the model, we encode each state $x$ as a one-hot vector in $\mathbb{R}^{2|E|}$, and we use a single-layer softmax predictor as follows:

$$\pi_\Theta(\cdot|x) = \text{softmax}(\langle \Theta, x \rangle), \quad \Theta \in (\mathbb{R} \cup \{-\infty\})^{2|E| \times 2|E|}.$$

We remark that this model is strictly more expressive than a *trigram model over nodes*: any trigram $\varphi(a|b, c)$ can be encoded as $\pi_\Theta(b \leftrightarrow c | a \leftrightarrow b)$, and moreover $\varphi$ cannot express multiedges.

Initially, the predictor learns the world model or underlying graph, which we view as the pretrained model; this viewpoint has been used to study pathfinding tasks by Kim et al. (2025). For a given edge $u \to v$, we set the transition probabilities to be $1/|N_v|$ to edges in $N_v$, and 0 otherwise. When $\Theta$ is clear from context, we will often omit the subscript and denote the policy simply as $\pi$. We will also denote the underlying distribution of the described world model as $\mathcal{D}$, from which we draw samples $(s, a)$ for current edge state and next edge state pairs such that the marginal probability over $s$ is uniform over all edge states.

**Theorem 1** (Warm-up: convergence of pretraining). *Suppose we train on the pairs $(s, a) \sim \mathcal{D}$ of current edge state and next edge state for our bigram such that $\mathbb{P}_{(s,a)\sim\mathcal{D}}[s]$ is uniform for all states. Under the loss*

$$L_{\text{pre}}(\Theta) := \mathbb{E}_{(s,a)\sim\mathcal{D}}[-\log \pi_\Theta(a|s)],$$

*it holds that standard gradient flow from zero initialization learns transition probabilities in finite time.*

The proof of Theorem 1 is deferred to Section A of the appendix. This justifies the following assumption on exactness of pretraining:

**Assumption 1.** *The pretrained model $\pi_{\Theta_{\text{pre}}}$ has converged arbitrarily close to the world model's distribution.*

This simplification is also used in Kim et al. (2025) to focus on analysis of post-training. Hence, $\pi_{\Theta_{\text{pre}}}$ can also be seen as a trigram over nodes, with multiedges taken into account.

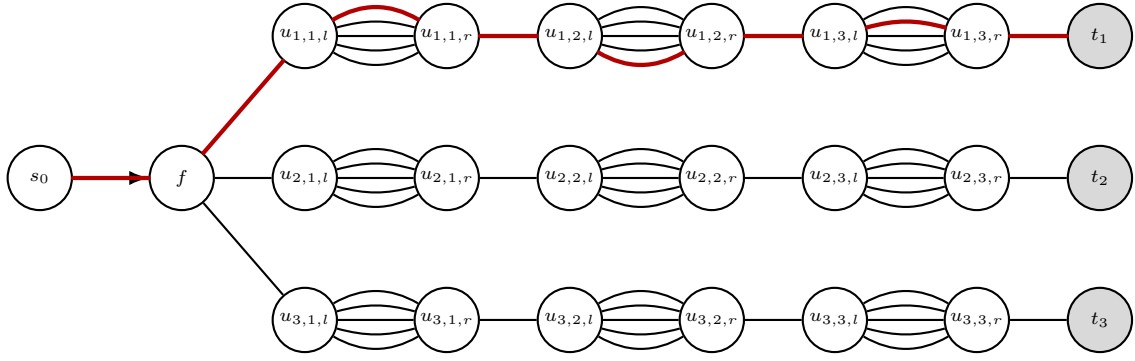

*Figure 1.* Structure of the world model graph, consisting of a source $s_0$, a fork $f$, and $W$ branches. Each branch contains a sequence of $K$ diamonds, each with $L$ multiedges, leading to a final leaf node. Here, $K = W = 3$ and $L = 5$. An example shortest-length path from $s_0$ to $t_1$ is highlighted in red.

### 3.3. Post-training

We study post-training $\pi_\Theta$ with either SFT or RLVR. In both setups, we consider training on data corresponding to path-target pairs, where the targets are sampled from the $W$ leaf nodes $t_i$. However, the main difference is that SFT is given these pairs, whereas RLVR must obtain them via on-policy rollouts. We detail these methods in the following sections. In both cases, we assume the generation does not depend on the prompted target. That is, it is a pure bigram or Markov chain with initial state $s \to f$ and transition probabilities $\pi_\Theta(\cdot)$.

#### 3.3.1. SFT

For SFT, we assume the model has access to golden shortest-length path examples, with no backtracking or exploration data. In practice, this corresponds to (for instance) feeding the model with gold standard solutions to a math problem. In particular, we train on shortest-length paths from the source $s_0$ to one of the $W$ leaf nodes $t_i$ for $1 \le i \le W$. These golden paths have the following characteristics: ((1) it chooses the correct branch at the fork node $f$, and (2) for each diamond $\Diamond(u_{i,j,l}, u_{i,j,r})$ for $1 \le j \le K$, it chooses exactly one of the $L$ multiedges to traverse.

We optimize the cross-entropy loss over the dataset of golden paths and targets:

$$\min_\Theta L(\Theta) = \mathbb{E}_{x \sim \mathcal{U}(t_i), y \sim \mathcal{D}_x} \left[ -\sum_{t=0}^{|y|-1} \log \pi_\Theta(y_{t+1}|y_t) \right] \tag{1}$$

where $\mathcal{D}_x$ is any (fully supported) distribution over golden shortest-length paths from the source to the target $x$. For ease of exposition, we consider optimizing this loss function via gradient flow, so that:

$$\frac{\mathrm{d}\Theta_{s,a}}{\mathrm{d}t} = -\frac{\partial L}{\partial \Theta_{s,a}}$$

for all pairs $s, a$ of current and next states, respectively.

#### 3.3.2. RLVR

For RLVR, we generate on-policy rollouts starting from the source $s_0$, with a verifier which checks whether the target node $t_i$ has been reached. As with many open-source models (Kimi et al., 2025; DeepSeek-AI et al., 2025b), we will employ the use of a length penalty along with the verifier outcome reward, which for pathfinding is simply the length of the rollout. That is, our loss function is

$$\max_\Theta J(\Theta) = \mathbb{E}_{x \sim \mathcal{U}(t_i), y \sim \pi_\Theta}[r(x, y)], \tag{2}$$

$$r(x, y) = \mathbf{1}\{y \text{ hits node } x\} - \beta|y|,$$

with an appropriately chosen length penalty $\beta$. Alternatively, we could choose to use a finite horizon instead of a length penalty and heuristically expect the same results.

For a given rollout with target $x = t_i$, we assume that the verifier stops the rollout as soon as the current state $s$ becomes $u_{i,K,r} \to x$. Then with a slight abuse of notation, $\mathbb{E}_{y \sim \pi}[|y|]$ represents the expected hitting time of $t_i$ (which is same for all targets by symmetry).

**Policy gradient update.** For the dynamical analysis, we consider the policy gradient update (Sutton et al., 1999):

$$\nabla_\Theta J(\Theta) = \mathbb{E}_x \mathbb{E}_{y \sim \pi}[r(x, y) \nabla_\Theta \log \pi_\Theta(y)]$$

$$= \mathbb{E}_x \mathbb{E}_{y \sim \pi} \left[ r(x, y) \sum_{t=0}^{|y|-1} \nabla_\Theta \log \pi_\Theta(s_{t+1}|s_t) \right]$$

where $s_t$ is the current edge state, $s_{t+1}$ is a potential next edge state, and the policy does not depend on the target.

Without loss of generality, we will work with the case where the length penalty strength $\beta = 1$. For simplicity of analysis, we consider policy gradient optimization via signed gradient

flow. That is,

$$\frac{\mathrm{d}\Theta_{s,a}}{\mathrm{d}t} = \mathrm{sgn}\left(\frac{\partial J}{\partial \Theta_{s,a}}\right)$$

for all pairs $s, a$ of current and next state, respectively. Signed gradient descent has also been studied in Kim et al. (2025) to avoid fringe non-convergence issues.

## 4. Main Results

### 4.1. Notation

To rigorously analyze the two algorithms, we first observe that by symmetry of the population gradient and the graph topology, the transition probabilities for "topologically equivalent" edge states on different branches (i.e., at the same depth and same orientation with respect to the fork) are identical. Thus for any given $j$, the transition probabilities at any time $t \geq 0$ are the same for the following state types:

1. Forward (resp. backward) diamond connector states $u_{i,j-1,r} \to u_{i,j,l}$ (resp. $u_{i,j,l} \to u_{i,j-1,r}$) for all $1 \leq i \leq W$.

2. Forward (resp. backward) diamond multiedge states $u_{i,j,l} \overset{(\ell)}{\to} u_{i,j,r}$ (resp. $u_{i,j,r} \overset{(\ell)}{\to} u_{i,j,l}$) for all $1 \leq i \leq W$ and $1 \leq \ell \leq L$.

Due to multiedge states being identical logit-wise regardless of the choice of target, we will treat those states as identical; transitions into those states will have equal probability by symmetry. Hence, we denote a single multiedge state $u_{i,j,l} \to u_{i,j,r}$ to be the aggregate of $u_{i,j,l} \overset{(\ell)}{\to} u_{i,j,r}$, and similarly for $u_{i,j,r} \to u_{i,j,l}$.

We also use the following notation for states; see Figure 2.

1. $R_{i,j}^+$: state $u_{i,j,l} \to u_{i,j,r}$ (arrived at the right diamond node from the left). The probability of the forward connector (same direction as edge state) is denoted $a_j$, and the reverse multiedges back across the diamond have total probability $1 - a_j$ ($\frac{1-a_j}{L}$ per edge).

2. $R_{i,j}^-$: state $u_{i,j+1,l} \to u_{i,j,r}$ (arrived at the right diamond node from the right). The total probability of the backward multiedges is $b_j$ ($\frac{b_j}{L}$ per edge), and the state back across the same edge has probability $1 - b_j$.

3. $L_{i,j}^+$: state $u_{i,j-1,r} \to u_{i,j,l}$ (arrived at the left diamond node from the left). The total probability of the forward multiedges is $c_j$, and the state back across the same edge has probability $1 - c_j$.

4. $L_{i,j}^-$: state $u_{i,j,r} \to u_{i,j,l}$ (arrived at the left diamond node from the right). The probability of the backward connector is $d_j$, and the multiedge states back across the diamond has total probability $1 - d_j$.

For the four types of states, we will essentially have two types of transitions: going forwards and going backwards. Note that over all $i$, the states of the same type and of same depth will have the same transition probabilities. Therefore, when the context is clear, we will often suppress the subscript $i$ in the notation, and analyze the two cases of when $i$ is the target branch versus when $i$ is not the target branch. Finally, there are two special states which never have their logits updated:

1. $u_{i,K,r} \to t_i$: leaf-incoming states for $1 \leq i \leq W$. Since $t_i$ is only adjacent to the node $u_{i,K,r}$, the state $u_{i,K,r} \to t_i$ deterministically transitions to $t_i \to u_{i,K,r}$ (unless $t_i$ is the target, in which case the verifier stops the RL model rollout).

2. $f_i$: fork-incoming states $u_{i,1,l} \to f$ for $1 \leq i \leq W$. We fix the transition probabilities of these states to be uniform over the $W$ edge states $f \to u_{i,1,l}$, including the same branch it came from.[1]

We call the next edge states that the transition probabilities $a_j, b_j, c_j, d_j$ above correspond to **desired** states; the other actions are called **undesired** states.

In particular, we have the following:

$$a_j = \frac{\exp\left(\Theta_{R_j^+, L_{j+1}^+}\right)}{\exp\left(\Theta_{R_j^+, L_{j+1}^+}\right) + L \exp\left(\Theta_{R_j^+, L_j^-}\right)},$$

$$b_j = \frac{L \exp\left(\Theta_{R_j^-, L_j^-}\right)}{L \exp\left(\Theta_{R_j^-, L_j^-}\right) + \exp\left(\Theta_{R_j^-, L_{j+1}^+}\right)},$$

$$c_j = \frac{L \exp\left(\Theta_{L_j^+, R_j^+}\right)}{L \exp\left(\Theta_{L_j^+, R_j^+}\right) + \exp\left(\Theta_{L_j^+, R_{j-1}^-}\right)},$$

$$d_j = \frac{\exp\left(\Theta_{L_j^-, R_{j-1}^-}\right)}{\exp\left(\Theta_{L_j^-, R_{j-1}^-}\right) + L \exp\left(\Theta_{L_j^-, R_j^+}\right)}.$$

For a given state $s$, let $a_1$ be its desired next state, and $a_0$ be its undesired next state. Then the pretrained model satisfies $\Theta_{s,a_1} = \Theta_{s,a_0} = 0$ and $\Theta_{s,a'} = -\infty$ for $a'$ not in the set

---

[1]If these logits are also updated, it can be shown that they will converge to the uniform distribution on the remaining $W - 1$ branches, discarding the branch it has exited from. Since this does not affect the final guarantees order-wise, we fix these logits for simplicity.

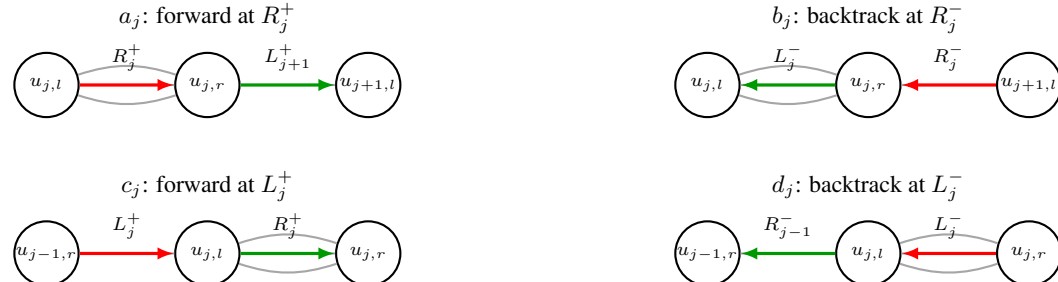

*Figure 2.* Desired edge-state transition types $a_j, b_j, c_j, d_j$. The current edge-state is denoted **red**; the desired next edge-state is denoted **green**.

of possible next actions, so that $a_j(0) = d_j(0) = \frac{1}{L+1}$ and $b_j(0) = c_j(0) = \frac{L}{L+1}$.

Finally, we denote $H_f(t) = \mathbb{E}_{y \sim \pi_{\Theta(t)}}[|y|]$ to be the expected hitting time at time $t$ after the start of post-training. Note that for any target $x = t_i$, this value is again the same.

### 4.2. SFT Learned Model

**Theorem 2** (Transitions learned by SFT). *When training with gradient flow on the SFT loss (Equation (1)) over only golden paths, it holds that $a_j, c_j$ converge arbitraily close to 1 in finite time. On the other hand, $b_j, d_j$ are fixed at $\frac{L}{L+1}, \frac{1}{L+1}$, respectively, over all time.*

Essentially, this theorem states that all forward-pointing edges will learn to put transition mass towards the next forward edge. On the other hand, because the model is only given golden paths towards the target, all backward pointing edge states remain uniform over its neighbors. In particular, because there is no explicit backtracking data in the fine-tuning dataset, the only ability to backtrack comes from what is learned during the pretraining phase, with no chance to be amplified post-training.

### 4.3. RLVR Learned Model

**Theorem 3** (Transitions learned by RLVR). *When training with sign gradient flow on the RLVR loss (Equation (2)), $a_j, b_j, c_j, d_j$ converge arbitrarily close to 1 in finite time.*

Intuitively, on-policy rollouts in RLVR penalizes the model for failed attempts containing long back-and-forth paths; it therefore needs to amplify its backtracking ability to mitigate this. As such, the learned model converges to the state where all edge states that point forward on a branch will learn to keep going forward, and all edge states that point backwards on a branch will learn to keep going backward.

### 4.4. Separation in Inference-time Efficiency

Given the policy learned from either SFT or RLVR, our main result shows that the former can be exponentially more

inefficient at inference time. This is due to the fact that fine-tuning on only the golden paths induces a lack of backtracking ability for the SFT model, which is well known to memorize the data distribution. For ease of exposition, we suppose that the SFT and RLVR models have converged to the transitions described in Theorem 2 and Theorem 3 (equivalently, the $t = \infty$ limit of the training process).

**Theorem 4** (Inference-time separation). *Suppose that the learned RLVR model with $a_j, b_j, c_j, d_j = 1$ and the learned SFT model are prompted with any target node $u$ in the graph to find a path starting from the source. Then the RLVR model requires $\Theta(WK)$ time in expectation to find $u$, while the SFT model requires $\Theta(WL^K)$ time.*

The above theorem implies a separation of $\Theta(L^K)$, which is exponential in the world model's depth.

In addition to the above separation result, it is also insightful to understand what happens in the case where we allow additional inference time compute to orchestrate the search, say using some search agent. Suppose the search agent is such that it keeps track of which edge states have been visited at inference time, and it will prevent the policy from entering a visited edge state. Intuitively, in the case of the RLVR-learned policy, there would be essentially no impact, since inside a given branch each directed edge visited will only be visited once (this is a property of the learned policy), and the agent's use would be purely in preventing it from visiting previously entered branches. For the SFT-learned policy, however, there will be an exponential improvement in inference-time efficiency, though not enough to match the RLVR policy. This is formalized in the following.

**Corollary 1.** *Given access to a search agent that prevents a given directed edge state from being visited twice during a single inference call, the RLVR model requires $\Theta(WK)$ time in expectation to find a target node $u$, while the SFT model requires $\Theta(WKL)$.*

*Proof.* To formalize the intuition above, note that the SFT policy will still take $\Theta(K)$ time to reach a leaf of a given branch (since it goes forward with probability one). In a

non-target branch, we claim that it will take $\Theta(KL)$ time to exit. This is because at the state $R_j^-$, we know that it will already have visited $L_{j+1}^+$ in the past; hence, it will take $\Theta(L)$ generations before it will continue to go backwards (e.g. $\Theta(L)$ rejected actions of going forwards by the search agent). Since there are $W$ branches, we accrue a factor of $W$ as well. $\qquad\square$

Indeed, there is still a $\Theta(L)$ factor separation between RLVR trained with backtracking against SFT on golden paths. Nevertheless, the fact that the search agent can allow an exponential speedup for some policies (albeit in this toy setting) demonstrates its power. Recent works have explored this axis of scaling test-time compute in many ways (Yao et al., 2024; Dang et al., 2026), and it is an open question to develop better orchestration and inference pipelines.

### 4.5. Distilling RLVR Reasoning Traces

Finally, we highlight when SFT can succeed for our reasoning task. Suppose we have access to the reasoning traces of a trained model, such as the outputs of a powerful closed-source model. If we fine-tune a pretrained base model on such traces (a process known as distillation (Shridhar et al., 2022)), we obtain similar guarantees in terms of inference-time efficiency, thereby avoiding the exponential blowup in Theorem 4.

**Theorem 5** (SFT on reasoning traces). *Denote the joint distribution of target and reasoning trace pairs generated by the RLVR converged model to be $\mathcal{D}$. Then, if we fine-tune the pretrained model on these traces by optimizing the following loss:*

$$\min_{\Theta} L_{\mathrm{distill}}(\Theta) = \mathbb{E}_{(x,y)\sim\mathcal{D}}\left[-\sum_{t=0}^{|y|-1} \log \pi_{\Theta}(y_{t+1}|y_t)\right]$$

*we can achieve $\Theta(WK)$ expected inference time compute.*

The fact that reasoning traces learned by a reinforcement fine-tuned model are useful for distillation demonstrates the importance of backtracking in fine-tuning data.

## 5. Overview of Proofs

### 5.1. Proof Sketch of SFT Training

We essentially consider initialization from the pretrained model and analyze the gradient update on the cross-entropy loss. Since only the logits corresponding to $a_j, c_j$ change (as the training data does not contain any states $R_j^-$ or $L_j^-$), we can reduce the argument into an analysis of an ODE on the logit gaps, and this is sufficient to prove Theorem 2.

### 5.2. Proof Sketch of RLVR Training

In contrast to the SFT setting, the dynamical analysis for RLVR is significantly more involved as the interaction between rollouts and backtracking comes into play. For each logit, the policy gradient is equivalent to

$$\frac{\partial J}{\partial \Theta_{s,a}} = \mathbb{E}_x[d_x(s)\pi(a|s)A_x(s,a)]$$

where $d_x(s) := \mathbb{E}_y[\sum_{t<\tau} \mathbf{1}\{s_t = s\}]$ is the expected number of times we reach state $s$ before hitting the target $x$, and $A_x(s,a)$ is the advantage for the policy $\pi$ for the reward (hitting time) when choosing action $a$ compared to the other actions at state $s$.

In our analysis of RLVR, this also means that we will treat the previously defined quantity $d_x(\cdot)$ for such states as:

$$d_x(u_{i,j,l} \to u_{i,j,r}) := \sum_{\ell=1}^{L} \mathbb{E}_y\left[\sum_{t<\tau} \mathbf{1}\{s_t = (u_{i,j,l} \overset{(\ell)}{\to} u_{i,j,r})\}\right].$$

The analogous statement holds true for the reverse multi-edges as well.

For a fixed target $x$, define the hitting time $\tau_x := \min\{t \geq 0 : \mathrm{head}(s_t) = x\}$. Also, let $h_x(s) := \mathbb{E}[\tau_x|s_0 = s]$, with absorption $h_x(s) = 0$ if $\mathrm{head}(s) = x$ and the Bellman equation:

$$h_x(s) = 1 + \sum_a \pi(a|s)h_x(a) \quad (\mathrm{head}(s) \neq x).$$

Then the advantage satisfies:

$$A_x(s,a) = \bar{h}_x(s) - h_x(a), \quad \bar{h}_x(s) := \sum_{a'} \pi(a'|s)h_x(a').$$

Also, recall the expected number of visits to a directed-edge state $s$ before hitting $x$ is

$$d_x(s) := \mathbb{E}\left[\sum_{t<\tau_x} \mathbf{1}\{s_t = s\}\right].$$

For multiedge states, we will treat this quantity as an aggregate of all $L$ multiedges. Now, the update for each logit becomes:

$$\frac{\partial J}{\partial \Theta_{s,a}} = \mathbb{E}_x[d_x(s)\pi(a|s)A_x(s,a)]$$

$$= \frac{1}{W}\sum_{i=1}^{W} d_{t_i}(s)\pi(a|s)\big(\bar{h}_{t_i}(s) - h_{t_i}(a)\big).$$

In each type of state, there is a desired action and an undesired action. Our goal is to show that the learned model converges to the state where all edge states that point forward on a branch will learn to keep going forward, and all

edge states that point backwards on a branch will learn to keep going backward. The update rule of the logit difference of these two actions are related via the following lemma, which is proved in Section B.1.

**Lemma 1.** *For a state $s$, let $\Theta_{s,1}$ correspond to the logit for one of the desired next edge states, and $\Theta_{s,0}$ correspond to the logit for one of the undesird next edge states, and let the logit gap $\mathcal{D}_s := \Theta_{s,1} - \Theta_{s,0}$. Then*

$$\frac{\mathrm{d}\mathcal{D}_s}{\mathrm{d}t} = \mathrm{sgn}(\mathbb{E}_x[d_x(s)(h_x(a_0) - h_x(a_1))]).$$

As a corollary, we can express the updates of the logit differences as follows, where we suppress the subscript $i$ for brevity. The following hold for the four types of states:

1. $R_j^+$ state: Denote the logit gap as $\mathcal{D}_j^{(a)}$. Then,

$$\frac{\mathrm{d}\mathcal{D}_j^{(a)}}{\mathrm{d}t} = \mathrm{sgn}\underbrace{\mathbb{E}_x\left[d_x(R_j^+) \cdot \left(h_x(L_j^-) - h_x(L_{j+1}^+)\right)\right]}_{=:G_j^{(a)}}.$$

2. $R_j^-$ state: Denote the logit gap as $\mathcal{D}_j^{(b)}$. Then,

$$\frac{\mathrm{d}\mathcal{D}_j^{(b)}}{\mathrm{d}t} = \mathrm{sgn}\underbrace{\mathbb{E}_x\left[d_x(R_j^-) \cdot \left(h_x(L_{j+1}^+) - h_x(L_j^-)\right)\right]}_{=:G_j^{(b)}}.$$

3. $L_j^+$ state: Denote the logit gap as $\mathcal{D}_j^{(c)}$. Then,

$$\frac{\mathrm{d}\mathcal{D}_j^{(c)}}{\mathrm{d}t} = \mathrm{sgn}\underbrace{\mathbb{E}_x\left[d_x(L_j^+) \cdot \left(h_x(R_{j-1}^-) - h_x(R_j^+)\right)\right]}_{=:G_j^{(c)}}.$$

4. $L_j^-$ state: Denote the logit gap as $\mathcal{D}_j^{(d)}$. Then,

$$\frac{\mathrm{d}\mathcal{D}_j^{(d)}}{\mathrm{d}t} = \mathrm{sgn}\underbrace{\mathbb{E}_x\left[d_x(L_j^-) \cdot \left(h_x(R_j^+) - h_x(R_{j-1}^-)\right)\right]}_{=:G_j^{(d)}}.$$

Note that the dynamics depend exactly on the signs of $G_j^{(p)}$ for $p \in \{a, b, c, d\}$. We divide our analysis into two phases: 1) the time until all of the logit gaps are increasing (i.e., $G_j^{(p)} > 0$ for all $1 \leq j \leq K$ and $p \in \{a, b, c, d\}$) and 2) convergence from said time towards 1.

**RLVR at post-training initialization.** Denote the start of RLVR as time $t = 0$. The following lemma characterizes the properties of the gradients at initialization.

**Lemma 2.** *At $t = 0$, it holds that $G_j^{(a)}, G_j^{(c)} > 0$. However, $G_j^{(b)}, G_j^{(d)}$ may be negative for depths $1 \leq l(W, K, L) \leq j \leq r(W, K, L) \leq K - 1$, where $l(\cdot), r(\cdot)$ depend on the specific values of $W, K, L$.*

The full proof is deferred to the appendix. Intuitively, this can be interpreted as the following. First, forward-oriented states (corresponding to $G_j^{(a)}, G_j^{(c)}$) want to continue forwards, as going backwards would be a waste of movement. Second, backward-oriented states (corresponding to $G_j^{(b)}, G_j^{(d)}$) near the end of a branch want to continue backwards, because they tend to believe that they came from a non-target leaf on the current branch. Third, backward-oriented states near the fork tend to put higher preference on the additional optionality of moving towards the fork as opposed to continuing forward on an uncertain branch. Finally, backward-oriented states in the middle are more "confused," in the sense that it is unclear if they are facing backwards because of backtracking or because of a U-turn from a forward-oriented state; this confusion should eventually be resolved as forward-oriented states converge to putting all mass forwards.

**RLVR Phase I.** We begin by showing the forward states' logits are increasing over all time; that is, we show that $G_j^{(a)}, G_j^{(c)}$ are always positive. As such, we will prove that once all the $G_j^{(b)}, G_j^{(d)}$ become positive, all four of these quantities will be increasing (and analyzed in Phase II). Hence, we will bound the time $T_{\text{meet}}$ that it takes us to reach Phase II. This is given by the following lemma.

**Lemma 3.** *There exists a time $T_{\text{meet}} = O\left(\frac{1}{K} \log \frac{H_0}{L}\right)$ such that for all $1 \leq j \leq K$ and all $p \in \{a, b, c, d\}$, $G_j^{(p)}(T_{\text{meet}}) > 0$.*

The proof of this fact relies on analyzing how $G_j^{(p)}$ evolves over this time. In particular, we can show theoretically and empirically that the middle interval of depths for which $G_j^{(b)}$ and $G_j^{(d)}$ is negative becomes shorter and shorter in Phase I, which is at the core of the proof of Lemma 3. Full details are deferred to the appendix.

**RLVR Phase II.** Continuing on, we enter the second phase of the training dynamics after time $T_{\text{meet}}$. In the setting of Lemma 3, we now consider the additional time needed after time $t = T_{\text{meet}}$ to reach the regime where $\min_j (\min\{a_j, b_j, c_j, d_j\}) \geq 1 - \kappa$ for some $\kappa \ll 1$, for which we prove the following guarantee.

**Lemma 4.** *Let $T' = \Theta\left(\log \frac{L}{\kappa}\right)$. Then, at time $t = T_{\text{meet}} + T'$, the following holds:*

$$\min\{a_j(t), b_j(t), c_j(t), d_j(t)\} \geq 1 - \kappa \quad \forall 1 \leq j \leq K.$$

The result of this phase follows from the self-reinforcing nature of the dynamics in this regime. Combining the results thus far gives Theorem 3. Full proofs can be found in Section B in the appendix.

### 5.3. Proof Sketch of Inference Time Separation

Given the construction of the converged model in Theorem 3, the model takes $\Theta(K)$ time to traverse any branch back-and-forth, and traverses $W$ branches on expectation before hitting the target branch. Therefore, the required inference time compute is $\Theta(WK)$ for the RLVR model.

For the converged SFT model in Theorem 2, the backtracking behavior is more complex. Upon entering a branch $i$, since $a_j = c_j = 1$ for all $j$, we reach the $t_i$ node in $\Theta(K)$ time. However, exiting the branch is more complicated. Starting from the state $R_{K+1}^+$ with head $t_i$, let $g_i$ denote the expected time of first entry into $R_{K+1-i}^-$, and $f_i$ denote the expected time of first entry into $L_{K+1-i}^-$. Then, we have the following recursion with base case $g_1 = 1$:

$$f_i = \frac{L+1}{L}g_i + \frac{2i-1}{L} + 1,$$
$$g_i = (L+1)f_{i-1} + (2i-2)L + 1.$$

We prove and analyze this recurrence in Section C of the appendix. In particular, the time needed from entry of a wrong branch to getting back to the fork state is $g_{K+1} = O(L^K)$, leading to a total compute of $\Theta(WL^K)$.

## 6. Experiments

In this section, we run experiments on our synthetic setup to validate our theory. We provide example plots for the case of $W = 15, K = 15, L = 5$, run under sign gradient descent with learning rate 0.01. Figure 3 shows that the final hitting time indeed converges to $\Theta(KW)$, as suggested by Section 4.4. Moreover, Figure 4 shows that all probabilities $a_j, b_j, c_j, d_j$ indeed converge to 1 with a rate (only) depending on their type. While not visually clear, we note that there is a very brief regime in the beginning in which the probabilities $b$ and $d$ are decreasing for middle depth nodes.

## 7. Conclusion

We highlight backtracking as a key capability for reasoning. We introduce a simple graph pathfinding model of chain-of-thought reasoning and give a dynamical comparison between supervised fine-tuning on shortest-path demonstrations and RL with verifiable rewards. We show that SFT on gold traces need not update backward-state transitions and can incur exponential search cost, whereas RLVR's on-policy rollouts provide signal to learn efficient backtracking from failed paths, yielding a provable inference-time compute separation. Finally, we show that RLVR-generated

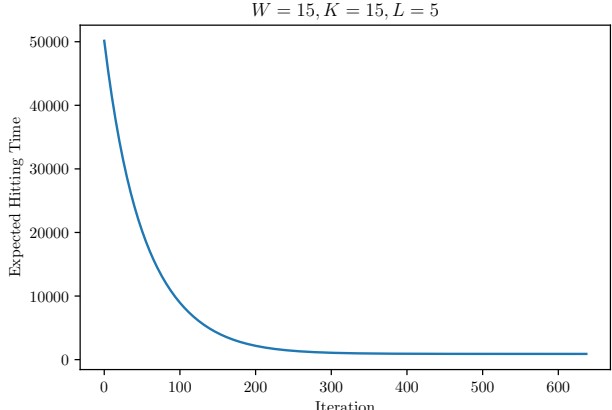

*Figure 3.* Expected hitting time of RLVR-trained model against training iterations. Here, hitting time converges to $4WK = 900$.

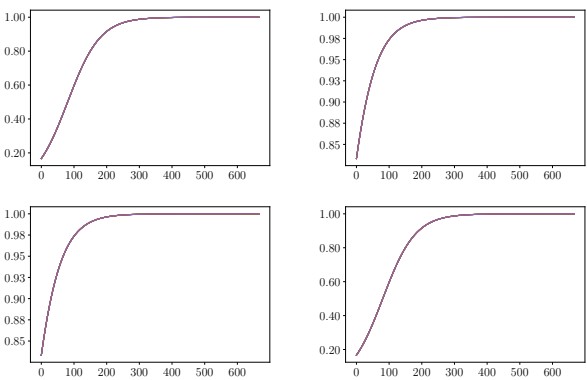

*Figure 4.* $a_j, b_j, c_j, d_j$ values for RLVR-trained model (ordered in Z shape). Values of edges over all depths $j$ are overlaid. Note that $a(0) = d(0) = \frac{1}{L+1}$ and $b(0) = c(0) = \frac{L}{L+1}$.

traces can be distilled via supervised learning to transfer efficient backtracking to a base model.

## Acknowledgements

SW acknowledges support from a NSF Graduate Research Fellowship.

## Impact Statement

This paper presents work whose goal is to advance the field of machine learning. There are many potential societal consequences of our work, none of which we feel must be specifically highlighted here.

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

# A. Pretraining and Supervised Fine-Tuning

We start off with the following general lemma for cross-entropy loss.

**Lemma 5.** *Let $\mathcal{Q}$ be any distribution over pairs $(s, a)$ of current edge-state $s$ and next edge-state $a$. Define the marginal $d_{\mathcal{Q}}(s) := \mathbb{P}_{(s,a)\sim\mathcal{Q}}[s]$ and the conditional $p_{\mathcal{Q}}(a \mid s) := \mathbb{P}_{(s,a)\sim\mathcal{Q}}[a \mid s]$. Consider the cross-entropy objective*

$$L_{\mathcal{Q}}(\Theta) := \mathbb{E}_{(s,a)\sim\mathcal{Q}}\big[-\log\pi_{\Theta}(a \mid s)\big].$$

*Then for every pair $(s, a)$,*

$$\frac{\partial L_{\mathcal{Q}}}{\partial \Theta_{s,a}} = d_{\mathcal{Q}}(s)\Big(\pi_{\Theta}(a \mid s) - p_{\mathcal{Q}}(a \mid s)\Big).$$

*In particular, under gradient flow $\frac{d\Theta_{s,a}}{dt} = -\partial L_{\mathcal{Q}}/\partial\Theta_{s,a}$, the dynamics of each row $\Theta_{s,\cdot}$ depends only on data transitions out of $s$; if $d_{\mathcal{Q}}(s) = 0$ then $\Theta_{s,\cdot}(t)$ remains constant.*

We now prove the pretraining convergence result.

*Proof of Theorem 1.* We first initialize $\Theta(0) = 0$ to be the zero logit matrix. Let $\mathcal{D}$ be the world-model transition distribution described in the main text. For each state $s = u \to v$, let $\mathcal{A}(s) := N_v$ denote the set of valid next edge-states and let $m_s := |\mathcal{A}(s)|$. Under $\mathcal{D}$ we have the conditional target distribution

$$p_{\mathcal{D}}(a \mid s) = \begin{cases} \frac{1}{m_s}, & a \in \mathcal{A}(s), \\ 0, & a \notin \mathcal{A}(s). \end{cases}$$

By Lemma 5, the pretraining gradient flow satisfies, for every $s, a$,

$$\frac{d\Theta_{s,a}}{dt} = -d_{\mathcal{D}}(s)\big(\pi_{\Theta(t)}(a \mid s) - p_{\mathcal{D}}(a \mid s)\big).$$

By assumption, we have that all of the states of the world model have equally likely marginal probability. Hence, for all of the gradient flow logits, we can rescale time by $d_{\mathcal{D}}(s)$, so that:

$$\frac{d\Theta_{s,a}}{dt} = -\big(\pi_{\Theta(t)}(a \mid s) - p_{\mathcal{D}}(a \mid s)\big).$$

We now fix any state $s$. Because $\Theta_{s,a}(0) = 0$ for all $a$, we have complete symmetry at $t = 0$. Moreover, the target distribution $p_{\mathcal{D}}(\cdot \mid s)$ is invariant under permutations within the two groups $\mathcal{A}(s)$ and $\mathcal{A}(s)^c$. Since the ODE is deterministic and coordinates within each group have identical right-hand sides whenever they are equal, uniqueness of ODE solutions implies that for all $t \geq 0$,

$$\Theta_{s,a}(t) = u_s(t) \ \forall a \in \mathcal{A}(s), \qquad \Theta_{s,a}(t) = v_s(t) \ \forall a \notin \mathcal{A}(s)$$

for some scalars $u_s(t), v_s(t)$.

Let $M := 2|E|$ be the total number of edge-states, and write $n_s := M - m_s$. Define the total probability mass assigned to valid actions

$$p_s(t) := \sum_{a\in\mathcal{A}(s)} \pi_{\Theta(t)}(a \mid s) = \frac{m_s e^{u_s(t)}}{m_s e^{u_s(t)} + n_s e^{v_s(t)}}.$$

Then each valid action has probability $p_s(t)/m_s$ and each invalid action has probability $(1 - p_s(t))/n_s$. Applying Lemma 5 to one representative valid action and one representative invalid action gives

$$\frac{du_s}{dt} = \left(\frac{1}{m_s} - \frac{p_s(t)}{m_s}\right) = \frac{1 - p_s(t)}{m_s}, \qquad \frac{dv_s}{dt} = -\frac{1 - p_s(t)}{n_s}.$$

Hence for the logit gap $g_s(t) := u_s(t) - v_s(t)$ we have

$$\frac{dg_s}{dt} = (1 - p_s(t))\left(\frac{1}{m_s} + \frac{1}{n_s}\right) \geq 0,$$

so $g_s(t)$ is nondecreasing. Writing $w_s(t) := e^{g_s(t)}$ and using

$$p_s(t) = \frac{m_s w_s(t)}{m_s w_s(t) + n_s} \quad \Rightarrow \quad 1 - p_s(t) = \frac{n_s}{m_s w_s(t) + n_s},$$

we obtain

$$\frac{\mathrm{d}w_s}{\mathrm{d}t} = w_s(t)\frac{\mathrm{d}g_s}{\mathrm{d}t} = \left(\frac{1}{m_s} + \frac{1}{n_s}\right)\frac{n_s w_s(t)}{m_s w_s(t) + n_s} \geq \frac{1}{m_s},$$

where for the final inequality we have used that $w_s(t) \geq w_s(0) = 1$ and thus $\frac{n_s w_s}{m_s w_s + n_s} \geq \frac{n_s}{m_s + n_s}$ to yield the cancellation. Therefore $w_s(t) \geq 1 + \frac{1}{m_s}t$ and hence

$$1 - p_s(t) = \frac{n_s}{m_s w_s(t) + n_s} \leq \frac{n_s}{m_s\left(1 + \frac{1}{m_s}t\right)} = \frac{n_s}{m_s + t}.$$

Thus the total invalid mass $1 - p_s(t)$ goes to 0 as $t \to \infty$, and the valid mass goes to 1. Because all valid logits remain equal, the distribution over valid actions is uniform at all times:

$$\pi_{\Theta(t)}(a \mid s) = \frac{p_s(t)}{m_s} \quad \forall a \in \mathcal{A}(s).$$

Consequently, for any $\varepsilon > 0$ we may choose $T_s(\varepsilon) := n_s/\varepsilon$ so that for all $t \geq T_s(\varepsilon)$ we have $1 - p_s(t) \leq \varepsilon$, and therefore

$$\max_{a \in \mathcal{A}(s)} \left|\pi_{\Theta(t)}(a \mid s) - \frac{1}{m_s}\right| \leq \frac{\varepsilon}{m_s} \quad \text{and} \quad \sum_{a \notin \mathcal{A}(s)} \pi_{\Theta(t)}(a \mid s) \leq \varepsilon.$$

Since this holds for every state $s$, pretraining under gradient flow learns the world-model transition kernel up to arbitrary error in finite time, as desired. $\qquad\square$

Using a similar technique of analyzing a training with a teacher distribution using cross-entropy loss, we can prove the convergence of the SFT model. In the following section, we provide the proof for Theorem 2.

*Proof of Theorem 2.* We assume the SFT phase is initialized at the pretrained world model described in the text, i.e., for every state $s$ all valid next-actions have equal logits, and invalid actions have zero probability (equivalently logits $-\infty$). We also assume $\mathcal{D}_x$ is uniform with respect to the targets; this only affects the relative timescales of convergence.

Let $\mathcal{Q}_{\mathrm{SFT}}$ denote the induced distribution over adjacent transition pairs $(s, a)$ obtained by the following procedure: sample a target $x \sim \mathcal{U}(\{t_i\})$, sample a golden path $y \sim \mathcal{D}_x$, and then sample a uniformly random transition $(y_{t-1}, y_t)$ (i.e. each edge connecting diamonds, and each of the $L$ multiedges at a diamond will be sampled uniformly at random).

Then the SFT objective can be written as

$$L_{\mathrm{SFT}}(\Theta) = \mathbb{E}_{(s,a) \sim \mathcal{Q}_{\mathrm{SFT}}}\left[-\log \pi_\Theta(a \mid s)\right].$$

By Lemma 5, for each state $s$ the gradient flow depends only on the conditional $p_{\mathcal{Q}_{\mathrm{SFT}}}(\cdot \mid s)$, and if a state $s$ is never encountered in golden paths then its row never updates.

First, observe that by construction, every golden path is a shortest path from $s_0$ to some $t_i$ and therefore never contains backtracking. In particular, none of the backward-pointing states $R_{i,j}^-$ or $L_{i,j}^-$ appear in $y$. Hence for each such backward state $s$ we have $d_{\mathcal{Q}_{\mathrm{SFT}}}(s) = 0$ and Lemma 5 implies $\Theta_{s,\cdot}(t)$ is constant over time. Therefore the associated transition probabilities remain equal to their pretrained values:

$$b_j(t) = b_j(0) = \frac{L}{L+1}, \qquad d_j(t) = d_j(0) = \frac{1}{L+1}.$$

We now analyze the forward-oriented states. First, consider a forward-oriented state $R_{i,j}^+$, each of which have exactly one forward action and $L$ backward actions. On any golden path, whenever we are in state $R_{i,j}^+$ the next state is deterministically

the forward connector state $L_{i,j+1}^+$. Therefore the SFT conditional satisfies $p_{\mathcal{Q}_{\mathrm{SFT}}}(\cdot \mid R_{i,j}^+)$ being a point mass on the forward connector action.

By symmetry among the $L$ backward multiedges and the symmetric pretrained initialization, their logits remain equal for all time, so the row can be summarized by two scalars: $u(t)$ (logit for the unique forward connector) and $v(t)$ (common logit for each backward multiedge). Then

$$a_j(t) = \frac{e^{u(t)}}{e^{u(t)} + Le^{v(t)}}.$$

Applying Lemma 5 to this row yields (up to time rescaling of $d_{\mathcal{Q}_{\mathrm{SFT}}}(R_{i,j}^+)$):

$$\frac{\mathrm{d}u}{\mathrm{d}t} = 1 - a_j(t), \qquad \frac{\mathrm{d}v}{\mathrm{d}t} = -\frac{1 - a_j(t)}{L}.$$

Hence for the logit gap $g(t) := u(t) - v(t)$ we have

$$\frac{\mathrm{d}g}{\mathrm{d}t} = (1 - a_j(t))\left(1 + \frac{1}{L}\right) \geq 0.$$

Let $w(t) := e^{g(t)}$. Using $a_j(t) = \frac{w(t)}{w(t)+L}$ one computes

$$\frac{\mathrm{d}w}{\mathrm{d}t} = w(t)\frac{\mathrm{d}g}{\mathrm{d}t} = (L+1)\frac{w(t)}{w(t)+L} \geq 1,$$

since $w(t) \geq w(0) = 1$. Therefore $w(t) \geq 1 + t$ and

$$1 - a_j(t) = \frac{L}{w(t)+L} \leq \frac{L}{t+L+1}.$$

Thus for any $\kappa > 0$, taking $t \geq \frac{L}{\kappa} - L - 1$ ensures $a_j(t) \geq 1 - \kappa$.

Finally, we consider a forward-oriented state of type $L_{i,j}^+$. This state has $L$ forward multiedges transitions and one backward transition. On any golden path, whenever we are at $L_{i,j}^+$ the next action is always one of the forward multiedges across $\Diamond(u_{i,j,l}, u_{i,j,r})$. Therefore the SFT conditional satisfies $p_{\mathcal{Q}_{\mathrm{SFT}}}(\cdot \mid L_{i,j}^+)$ having zero probability on the backward transition.

Let $\rho_j(t)$ denote the model's probability of taking the backward connector at $L_{i,j}^+$; then $c_j(t) = 1 - \rho_j(t)$. The backward connector logit is pushed down at rate proportional to $\rho_j(t)$, and hence $\rho_j(t) \to 0$ and $c_j(t) \to 1$. Under the additional symmetry assumption that $\mathcal{D}_x$ is invariant under permutations of the $L$ multiedges in each diamond (so each forward multiedge is equally likely in the population loss), the same reduction as above applies: all $L$ forward multiedges share a common logit $u(t)$ and the backward connector has logit $v(t)$, so

$$c_j(t) = \frac{Le^{u(t)}}{Le^{u(t)} + e^{v(t)}} = \frac{Lw(t)}{Lw(t) + 1}, \quad w(t) := e^{u(t)-v(t)}.$$

The same calculation gives $\frac{\mathrm{d}w}{\mathrm{d}t} \geq 1/L$, hence $w(t) \geq 1 + t/L$ and

$$1 - c_j(t) = \frac{1}{Lw(t)+1} \leq \frac{1}{t+L+1}.$$

Thus for any $\kappa > 0$, taking $t \geq \frac{1}{\kappa} - L - 1$ ensures $c_j(t) \geq 1 - \kappa$, as desired. $\qquad\square$

## B. RLVR

### B.1. Setup and Notations

Recall that:

$$\nabla_\Theta J(\Theta) = \mathbb{E}_x \mathbb{E}_{y\sim\pi}[r(x,y)\nabla_\Theta \log \pi_\Theta(y)] = \mathbb{E}_x \mathbb{E}_{y\sim\pi_\Theta}\left[r(x,y)\sum_{t=0}^{|y|-1}\nabla_\Theta \log \pi_\Theta(s_{t+1}|s_t)\right].$$

In the main text, we claimed that this is equivalent to:

$$\frac{\partial J}{\partial \Theta_{s,a}} = \mathbb{E}_x[d_x(s)\pi(a|s)A_x(s,a)]$$

where $d_x(s) := \mathbb{E}_y[\sum_{t<\tau} \mathbf{1}\{s_t = s\}]$ is the expected number of times we reach state $s$ before hitting the target $x$, and $A_x(s,a)$ is the advantage for the policy $\pi$ for the reward (hitting time) when choosing action $a$ compared to the other actions at state $s$. We prove this claim below.

**Lemma 6.** *The policy gradient can be expressed as:*

$$\mathbb{E}_x\mathbb{E}_{y\sim\pi}\left[r(x,y)\sum_{t=0}^{|y|-1}\nabla_\Theta \log \pi(s_{t+1}|s_t)\right] = \mathbb{E}_x[d_x(s)\pi(a|s)A_x(s,a)]$$

*where*

$$A_x(s,a) = \bar{h}_x(s) - h_x(a) \quad \text{where} \quad \bar{h}_x(s) := \sum_{a'}\pi(a'|s)h_x(a').$$

*Proof.* First, we note that under softmax parameterization, we have

$$\pi_\Theta(a|s) = \frac{\exp(\Theta_{s,a})}{\sum_{a'}\exp(\Theta_{s,a'})}$$

and so

$$\frac{\partial \log \pi_\Theta(a_t|s_s)}{\partial \Theta_{s,a}} = \mathbf{1}\{s_t = s\}(\mathbf{1}\{a_t = a\} - \pi_\Theta(a|s)).$$

In our case, we have that $r(x,y) = 1 - |y|$. We can first simplify by noting that for any fixed $t$,

$$\mathbb{E}[\nabla_\Theta \log \pi_\Theta(a_t|s_t)|s_t] = \sum_a \pi_\Theta(a|s_t)\nabla_\Theta(a|s_t) = \nabla_\Theta\left(\sum_a \pi_\Theta(a|s_t)\right) = 0.$$

Therefore, we obtain:

$$\nabla_\Theta J(\Theta) = \mathbb{E}_{x,y}\left[\sum_{t=0}^{|y|-1}(1-|y|)\mathbf{1}\{s_t = s\}(\mathbf{1}\{a_t = a\} - \pi_\Theta(a|s))\right]$$
$$= -\mathbb{E}_{x,y}\left[\sum_{t=0}^{|y|-1}|y|\mathbf{1}\{s_t = s\}(\mathbf{1}\{a_t = a\} - \pi_\Theta(a|s))\right]$$

To continue, we note that for any fixed $t$, it holds that:

$$\mathbb{E}[t\nabla_\Theta \log_\Theta(a|s_t)|s_t] = 0$$

due to a similar argument as before (rewards from the past do not impact the gradient). Therefore, we can rewrite the policy gradient as:

$$\nabla_\Theta J(\Theta) = -\mathbb{E}_{x,y}\left[\sum_{t=0}^{|y|-1}|y|\mathbf{1}\{s_t = s\}(\mathbf{1}\{a_t = a\} - \pi_\Theta(a|s))\right]$$
$$= -\mathbb{E}_{x,y}\left[\sum_{t=0}^{|y|-1}(|y|-t)\mathbf{1}\{s_t = s\}(\mathbf{1}\{a_t = a\} - \pi_\Theta(a|s))\right]$$
$$= \mathbb{E}_x\left[d_x(s)\pi(a|s)(\bar{h}_x(s) - h_x(a))\right]$$

as desired. □

We are now ready to prove Lemma 1.

*Proof of Lemma 1.* First, we note that $\pi(a_0|s) + \pi(a_1|s) = 1$. Then, we observe that:

$$\bar{h}_x(s) - h_x(a_1) = \pi(a_0|s)h_x(a_0) + \pi(a_1|s)h_x(a_1) - h_x(a_1)$$
$$= \pi(a_0|s)(h_x(a_0) - h_x(a_1))$$

Similarly,

$$\bar{h}_x(s) - h_x(a_0) = -\pi(a_1|s)(h_x(a_0) - h_x(a_1))$$

Observe that $\bar{h}_x(s) - h_x(a_1)$ and $\bar{h}_x(s) - h_x(a_0)$ have opposite signs. Combining with Lemma 6, we obtain:

$$\frac{\mathrm{d}\mathcal{D}_s}{\mathrm{d}t} = \mathrm{sgn}\big(\mathbb{E}_x\big[d_x(s)\big[\pi(a_1|s)(\bar{h}_x(s) - h_x(a_1)) - \pi(a_0|s)(\bar{h}_x(s) - h_x(a_0))\big]\big]\big)$$
$$= \mathrm{sgn}(2\mathbb{E}_x[d_x(s)\pi(a_1|s)\pi(a_0|s)(h_x(a_0) - h_x(a_1))])$$
$$= \mathrm{sgn}(\mathbb{E}_x[d_x(s)(h_x(a_0) - h_x(a_1))])$$

as desired. □

To characterize the dynamics of RLVR, we first define the following notations.

**Definition 1.** *Fix a target $x$. We define the following quantities:*

1. *Let $H_f := h_x(u_{i,1,l} \to f)$ denote the hitting time of a fixed target $x$ from any of the states whose head is the fork $f$ (including the initial source state $s_0 \to f$); by symmetry they are all equal, hence why we consider a single value.*

2. *For state $s$ not on the branch of $x$, we define $\tilde{g}(s)$ to be the expected time of hitting the fork, starting at state $s$. That is, $\tilde{g}(s) := \mathbb{E}[\tau_f]$ where $\tau_f := \min\{t \geq 0 : s_0 = s, \mathrm{head}(s_t) = f\}$. In particular, this means that $h_x(s) = \tilde{g}(s) + H_f$, and we have the absorbing condition $h_x(u_{i,1,l} \to f) = 0$ for branch $i$ not equal to the branch of target $x$.*

3. *For state $s$ on the branch of $x$, we define $g(s)$ to be the expected time of hitting either $f$ or $x$ (e.g. both states are absorbing). That is, $g(s) := \mathbb{E}[\tau_{\{f,x\}}]$ where $\tau_{\{f,x\}} := \min\{t \geq 0 : s_0 = s, \mathrm{head} \in \{f,x\}\}$.*

4. *For state $s$ on the branch of $x$, we define $q(s)$ to be the probability it hits $f$ before $x$. In particular, this means that $h_x(s) = g(s) + q(s)H_f$.*

**Lemma 7.** *Define the following q-gaps:*

$$\delta_j := q(L_j^-) - q(L_{j+1}^+), \quad \epsilon_j := q(R_{j-1}^-) - q(R_j^+)$$

*Then, it holds that $\delta_j > 0$ and $\epsilon_j > 0$ for all $j$.*

*Proof.* For $\delta_j$, this follows intuitively from the fact that all paths from $L_{j+1}^+$ to the fork must pass through $L_j^-$. Hence, the event that a path arrives back at the fork from $L_{j+1}^+$ is a subset of the event that a path arrives back to the fork from $L_j^-$. Similar reasoning holds for $\epsilon_j$, and the lemma follows. □

**Lemma 8.** *Define the following $\tilde{g}$-gaps:*

$$\Delta_j := \tilde{g}(L_{j+1}^+) - \tilde{g}(L_j^-), \quad E_j := \tilde{g}(R_j^+) - \tilde{g}(R_{j-1}^-)$$

*Then, it holds that $\Delta_j \geq 2$ and $E_j \geq 2$ for all $j$.*

*Proof.* For $\Delta_j$, this follows from intuitively from the fact that all paths from $L_{j+1}^+$ to the fork must pass through $L_j^-$, which requires at least two moves. A similar reasoning holds for $E_j$, and the lemma follows. □

We remark that we cannot immediately obtain clean positivity results for the $g$-gaps (i.e., the expected absorbing time onto $\{f,x\}$ on for states on the branch of a target $x$). To analyze it, we first define the following quantities.

**Definition 2.** *Fix a target $x$, and consider the entry state $L_1^+$ upon entering any branch from the fork. We define the following for a state $s$ on this same branch:*

1. *When $s$ is on the target branch, define $\mu(s) := \mathbb{E}\left[\sum_{t<\tau_{\{f,x\}}} \mathbf{1}\{s_t = s\}|s_0 = L_1^+\right]$. In other words, $\mu_s$ is the expected number of visits to state $s$ during a target branch attempt.*

2. *When $s$ is not on the target branch, define $\tilde{\mu}(s) := \mathbb{E}\left[\sum_{t<\tau_f} \mathbf{1}\{s_t = s\}|s_0 = L_1^+\right]$. In other words, $\tilde{\mu}_s$ is the expected number of visits to state $s$ during a non-target branch attempt until it goes back to $f$.*

**Definition 3.** *Fix a target $x$, and define the success probability of hitting the target upon entering its branch $p_{\text{succ}}$ before hitting $f$. That is,*

$$p_{\text{succ}} := \mathbb{P}\{\text{head}(s_{\tau_{\{f,x\}}}) = x|s_0 = L_1^+\}$$

*where $L_1^+$ is the entry state of the branch of $x$.*

**Proposition 1.** *Fix a target $x$. Then, the following hold:*

1. *If $s$ is on the same branch as $x$, then we have $d_x(s) = \frac{\mu(s)}{p_{\text{succ}}}$.*

2. *If $s$ is on a different branch from $x$, then we have $d_x(s) = \frac{\tilde{\mu}(s)}{p_{\text{succ}}}$.*

*Proof.* Note that the probability of success for each branch entry is $\frac{p_{\text{succ}}}{W}$, as one needs to choose the target branch with probability $1/W$ and then succeed with probability $p_{\text{succ}}$. Hence, the total number of fork departures on expectation is $W/p_{\text{succ}}$, so the expected number of entries into any fixed branch is $1/p_{\text{succ}}$. $\square$

**Proposition 2.** *Define:*

$$\Delta g_j^{(a)} := g(L_j^-) - g(L_{j+1}^+), \quad \Delta g_j^{(c)} := g(R_{j-1}^-) - g(R_j^+)$$

*Furthermore, define $\Delta g_j^{(b)} := -\Delta g_j^{(a)}$ and $\Delta g_j^{(d)} := -\Delta g_j^{(c)}$. Then, the following hold:*

1. $G_j^{(a)} = \mathbb{E}_x\left[d_x(R_{i,j}^+) \cdot (h_x(L_{i,j}^-) - h_x(L_{i,j+1}^+))\right] = \frac{1}{Wp_{\text{succ}}}\left[\mu(R_j^+)\left(\Delta g_j^{(a)} + \delta_j H_f\right) - (W-1)\tilde{\mu}(R_j^+)\Delta_j\right]$

2. $G_j^{(b)} = \mathbb{E}_x\left[d_x(R_{i,j}^-) \cdot (h_x(L_{i,j+1}^+) - h_x(L_{i,j}^-))\right] = \frac{1}{Wp_{\text{succ}}}\left[\mu(R_j^-)\left(-\Delta g_j^{(a)} - \delta_j H_f\right) + (W-1)\tilde{\mu}(R_j^-)\Delta_j\right]$

3. $G_j^{(c)} = \mathbb{E}_x\left[d_x(L_{i,j}^+) \cdot (h_x(R_{i,j-1}^-) - h_x(R_{i,j}^+))\right] = \frac{1}{Wp_{\text{succ}}}\left[\mu(L_j^+)\left(\Delta g_j^{(c)} + \epsilon_j H_f\right) - (W-1)\tilde{\mu}(L_j^+)E_j\right]$

4. $G_j^{(d)} = \mathbb{E}_x\left[d_x(L_{i,j}^-) \cdot (h_x(R_{i,j}^+) - h_x(R_{i,j-1}^-))\right] = \frac{1}{Wp_{\text{succ}}}\left[\mu(L_j^-)\left(-\Delta g_j^{(c)} - \epsilon_j H_f\right) + (W-1)\tilde{\mu}(L_j^-)E_j\right]$

*Proof.* We will first consider the case $G_j^{(a)}$. If $R_j^+$ is on the target branch (which happens with probability $1/W$), then we have:

$$d_x(R_j^+) \cdot (h_x(L_{i,j}^-) - h_x(L_{i,j+1}^+)) = \frac{\mu(R_j^+)}{p_{\text{succ}}} \cdot \left((g(L_j^-) - g(L_{j+1}^+)) + (q(L_j^-) - q(L_{j+1}^+))H_f\right)$$

$$= \frac{\mu(R_j^+)}{p_{\text{succ}}} \cdot \left(\Delta g_j^{(a)} + \delta_j H_f\right)$$

If $R_j^+$ is not on the target branch (which happens with probability $(W-1)/W$), then we have:

$$d_x(R_j^+) \cdot (h_x(L_{i,j}^-) - h_x(L_{i,j+1}^+)) = \frac{\tilde{\mu}(R_j^+)}{p_{\text{succ}}} \cdot (\tilde{g}(L_j^-) - \tilde{g}(L_{j+1}^+)) = \frac{\tilde{\mu}(R_j^+)}{p_{\text{succ}}} \cdot (-\Delta_j)$$

This gives the desired expectation. Analogous calculations can be done for the other three cases to obtain the same result. $\square$

We now observe that we can rescale time in the four logit ODE's by a factor of $\frac{2}{W p_{\text{succ}}}$, since those terms do not change the sign in our gradient flow. Therefore, we redefine:

1. $G_j^{(a)} := \left[ \mu(R_j^+)\left(\Delta g_j^{(a)} + \delta_j H_f\right) - (W-1)\tilde{\mu}(R_j^+)\Delta_j \right]$

2. $G_j^{(b)} := \left[ \mu(R_j^-)\left(-\Delta g_j^{(a)} - \delta_j H_f\right) + (W-1)\tilde{\mu}(R_j^-)\Delta_j \right]$

3. $G_j^{(c)} := \left[ \mu(L_j^+)\left(\Delta g_j^{(c)} + \epsilon_j H_f\right) - (W-1)\tilde{\mu}(L_j^+)E_j \right]$

4. $G_j^{(d)} := \left[ \mu(L_j^-)\left(-\Delta g_j^{(c)} - \epsilon_j H_f\right) + (W-1)\tilde{\mu}(L_j^-)E_j \right]$

so that for $p_j \in \{a_j, b_j, c_j, d_j\}$, the gradient flow is

$$\frac{\mathrm{d}\mathcal{D}_j^{(p)}}{\mathrm{d}t} = \text{sgn}\left(G_j^{(p)}\right).$$

## B.2. Summary of Important Quantities

For ease of exposition, we summarize the notations defined above in a concise list with its correspondence to branch type.

- Off-target branch: $\tilde{\mu}(\cdot), \tilde{g}(\cdot), \Delta_j, E_j$

- On-target branch: $\mu(\cdot), g(\cdot), \Delta g_j^a, \Delta g_j^c, q(\cdot), \delta_j, \epsilon_j, p_{\text{succ}}$

- Overall: $H_f$, which is the expected hitting time of fixed target $t_i$ from a branch. Since the targets $t_i$ are symmetric, we have that $J(\Theta) = 1 - H_f$. Moreover, using Proposition 1 yields $H_f = \frac{W + g(L_1^+) + (W-1)\tilde{g}(L_1^+)}{p_{\text{succ}}}$.

Note that in the context of the population loss, the expectation is uniform over the choice of on target branch. In the remainder of this section, we will calculate these quantities exactly in terms of arbitrary $a_j, b_j, c_j, d_j$.

We give closed form expressions for all of the below quantities, which were calculated with the assistance of SymPy. We verify separately that these quantities (which are by definition unique) indeed hold for the stochastic system we have defined thus far. Unless denoted otherwise, the quantities below hold for all $1 \leq j \leq K$, with the convention that a state $R_0^-$ is a fork-head state, and $L_{K+1}^+$ is a leaf-head state, as well as the summation and product of an empty set being 0 and 1 respectively.

### B.2.1. Off-target quantities

Define the quantity $r_j := \frac{a_j c_j}{b_j d_j}$.

1. $\tilde{\mu}(\cdot)$: We have

$$\tilde{\mu}(L_j^+) = \prod_{m=1}^{j-1} r_m,$$

$$\tilde{\mu}(R_j^+) = \tilde{\mu}(L_j^-) = \frac{c_j}{d_j} \prod_{m=1}^{j-1} r_m,$$

$$\tilde{\mu}(R_j^-) = \tilde{\mu}(L_{j+1}^+) = \prod_{m=1}^{j} r_m.$$

2. $E_j$: We have

$$E_j = \sum_{m=j}^{K} \left( \frac{2(a_m + b_m)}{b_m d_m} \prod_{i=j}^{m-1} \frac{a_i c_{i+1}}{b_i d_i} \right).$$

Note the boundary convention $E_{K+1} := 0$.

3. $\Delta_j$: We have

$$\Delta_j = \frac{2 + c_{j+1}E_{j+1}}{b_j} = \frac{d_j E_j - 2}{a_j}.$$

4. $\tilde{g}(\cdot)$: Define $F_j := \tilde{g}(R_j^-)$, so that $F_0 = 0$. Then we have

$$F_j = \sum_{m=1}^{j} (2 + (1 - d_m)E_m + (1 - b_m)\Delta_m)$$

and

$$\tilde{g}(R_j^-) = F_j,$$
$$\tilde{g}(R_j^+) = F_{j-1} + E_j,$$
$$\tilde{g}(L_j^+) = 1 + F_{j-1} + c_j E_j,$$
$$\tilde{g}(L_j^-) = 1 + F_{j-1} + (1 - d_j)E_j.$$

We will generally not be working with $\tilde{g}$ directly, but rather their adjacent differences $E_j, \Delta_j$.

### B.2.2. ON-TARGET QUANTITIES

Define the following quantities:

$$r_j := \frac{a_j c_j}{b_{j-1} d_j}, \quad \forall\, 2 \le j \le K,$$

$$P_j := \prod_{m=j+1}^{K} r_m,$$

$$\alpha_j := (1 - a_j) + \frac{a_j}{d_j}(1 - c_j),$$

$$S_j := \sum_{m=j}^{K} \alpha_m P_m,$$

$$Z_q := P_1\left(1 - a_1 + \frac{a_1}{d_1}\right) + S_2.$$

We also have the following quantities.

1. $\delta_j$: We have $\delta_j = P_j/Z_q$.

2. $\epsilon_j$: We have $\epsilon_j = \delta_j a_j/d_j$.

3. $p_{\text{succ}}$: We have

$$p_{\text{succ}} = \frac{a_1 c_1}{d_1}\frac{P_1}{Z_q}.$$

4. $\Delta g_j^a$: First, define the following:

$$\mathcal{B}_j := -\sum_{m=j+1}^{K} \frac{2(c_m + d_m)}{b_{m-1}d_m} \prod_{i=j+1}^{m-1} r_i,$$

$$\beta_m := (1 - a_m) + \frac{a_m(1 - c_m)}{d_m},$$

$$\gamma_m := 2 - \frac{2(1 - c_m)}{d_m},$$

$$P_g := \sum_{m=2}^{K} \beta_m P_m,$$

$$Q_g := \sum_{m=2}^{K} (\beta_m \mathcal{B}_m + \gamma_m).$$

Then, it holds that:

$$\Delta g_K^a = -\frac{\left(\frac{a_1}{d_1} + 1 - a_1\right)\mathcal{B}_1 - \frac{2}{d_1} + Q_g + 1}{\left(\frac{a_1}{d_1} + 1 - a_1\right)P_1 + P_g},$$

$$\Delta g_j^a = P_j \Delta g_K^a + \mathcal{B}_j.$$

5. $\Delta g_j^c$: We have

$$\Delta g_j^c = \frac{a_j \Delta g_j^a - 2}{d_j}.$$

Equivalently, when $2 \leq j \leq K$,

$$\Delta g_j^c = \frac{b_{j-1}\Delta g_{j-1}^a + 2}{c_j}.$$

6. $g(\cdot)$: For simplicity, we will not write down the full expression, since we will generally be working with adjacent differences $\Delta g_j^a$ and $\Delta g_j^c$.

7. $\mu(\cdot)$: We define the prefix product $A_j := \prod_{m=1}^{j} a_m$, as well as $B_j, C_j, D_j$ similarly. Then,

$$\mu(L_j^+) = \frac{A_{j-1}C_{j-1}}{B_{j-1}D_{j-1}} \cdot \frac{S_j + \frac{a_j c_j}{d_j}P_j}{Z_q},$$

$$\mu(R_j^+) = \frac{A_{j-1}C_j}{B_{j-1}D_j} \cdot \frac{S_{j+1} + P_j}{Z_q},$$

$$\mu(L_j^-) = \mu(R_j^+) - p_{\text{succ}},$$

$$\mu(R_j^-) = \mu(L_{j+1}^+) - p_{\text{succ}}.$$

In the last line, for $j = K$ we use the auxiliary convention $\mu(L_{K+1}^+) := p_{\text{succ}}$, so that $\mu(R_K^-) = 0$.

### B.2.3. CLOSED FORMS FOR $G_j^{(p)}$

The aforementioned quantities are also sufficient for us to write a closed form for the rescaled $G_j^{(p)}$ for $p \in \{a, b, c, d\}$ used in the sign dynamics above. As before, define $A_j = \prod_{m=1}^{j} a_m$ to be the prefix product of $a$, and similarly for $B_j, C_j, D_j$.

In particular, we have:

$$G_j^{(a)} = \frac{A_{j-1}C_j}{B_{j-1}D_j} \cdot \frac{S_{j+1} + P_j}{Z_q} \left[ P_j \Delta g_K^a + \mathcal{B}_j + \frac{P_j}{Z_q} H_f \right] - (W - 1) \frac{A_{j-1}C_j}{B_{j-1}D_j} \Delta_j,$$

$$G_j^{(b)} = \left[ \frac{A_j C_j}{B_j D_j} \cdot \frac{S_{j+1} + b_j P_j}{Z_q} - p_{\text{succ}} \right] \cdot \left[ -P_j \Delta g_K^a - \mathcal{B}_j - \frac{P_j}{Z_q} H_f \right] + (W - 1) \frac{A_j C_j}{B_j D_j} \Delta_j,$$

$$G_j^{(c)} = \left[ \frac{A_{j-1}C_{j-1}}{B_{j-1}D_{j-1}} \cdot \frac{S_j + \frac{a_j c_j}{d_j} P_j}{Z_q} \right] \cdot \left[ \frac{a_j (P_j \Delta g_K^a + \mathcal{B}_j) - 2}{d_j} + \frac{a_j}{d_j} \frac{P_j}{Z_q} H_f \right] - (W - 1) \frac{A_{j-1}C_{j-1}}{B_{j-1}D_{j-1}} E_j,$$

$$G_j^{(d)} = \left[ \frac{A_{j-1}C_j}{B_{j-1}D_j} \cdot \frac{S_{j+1} + P_j}{Z_q} - p_{\text{succ}} \right] \cdot \left[ -\frac{a_j (P_j \Delta g_K^a + \mathcal{B}_j) - 2}{d_j} - \frac{a_j}{d_j} \frac{P_j}{Z_q} H_f \right] + (W - 1) \frac{A_{j-1}C_j}{B_{j-1}D_j} E_j.$$

### B.2.4. VERIFICATION OF THE CLOSED FORMS

We now verify that the displayed formulas solve the relevant linear systems. Since each system is finite and absorbing (i.e. from every transient state there is a fixed-length path to either the fork or the target with positive probability), we have that the Bellman and state transition equations have unique solutions. As such, we can verify the above closed form expressions via direct substitution and calculation.

We first list the equations to be checked. For off-target visit counts, the equations are

$$\tilde{\mu}(L_1^+) = 1,$$
$$\tilde{\mu}(L_{j+1}^+) = a_j \tilde{\mu}(R_j^+) + (1 - b_j)\tilde{\mu}(R_j^-),$$
$$\tilde{\mu}(R_j^+) = c_j \tilde{\mu}(L_j^+) + (1 - d_j)\tilde{\mu}(L_j^-),$$
$$\tilde{\mu}(L_j^-) = (1 - a_j)\tilde{\mu}(R_j^+) + b_j \tilde{\mu}(R_j^-),$$
$$\tilde{\mu}(R_j^-) = (1 - c_{j+1})\tilde{\mu}(L_{j+1}^+) + d_{j+1}\tilde{\mu}(L_{j+1}^-),$$

where for $j = K$ we use the leaf convention that the nonexistent state has $\tilde{\mu}(L_{K+1}^-) = 0$ and $L_{K+1}^+$ transitions to $R_K^-$ with probability one. For off-target hitting times, the Bellman equations are

$$\tilde{g}(R_0^-) = 0, \qquad \tilde{g}(L_{K+1}^+) = 1 + \tilde{g}(R_K^-),$$
$$\tilde{g}(L_j^+) = 1 + c_j \tilde{g}(R_j^+) + (1 - c_j)\tilde{g}(R_{j-1}^-),$$
$$\tilde{g}(L_j^-) = 1 + d_j \tilde{g}(R_{j-1}^-) + (1 - d_j)\tilde{g}(R_j^+),$$
$$\tilde{g}(R_j^+) = 1 + a_j \tilde{g}(L_{j+1}^+) + (1 - a_j)\tilde{g}(L_j^-),$$
$$\tilde{g}(R_j^-) = 1 + b_j \tilde{g}(L_j^-) + (1 - b_j)\tilde{g}(L_{j+1}^+).$$

On the target branch, the Bellman equations for $q$ have boundaries $q(R_0^-) = 1$ and $q(L_{K+1}^+) = 0$ and satisfy

$$q(L_j^+) = c_j q(R_j^+) + (1 - c_j)q(R_{j-1}^-),$$
$$q(L_j^-) = d_j q(R_{j-1}^-) + (1 - d_j)q(R_j^+),$$
$$q(R_j^+) = a_j q(L_{j+1}^+) + (1 - a_j)q(L_j^-),$$
$$q(R_j^-) = b_j q(L_j^-) + (1 - b_j)q(L_{j+1}^+).$$

The Bellman equations for $g$ have boundaries $g(R_0^-) = g(L_{K+1}^+) = 0$ and satisfy

$$g(L_j^+) = 1 + c_j g(R_j^+) + (1 - c_j)g(R_{j-1}^-),$$
$$g(L_j^-) = 1 + d_j g(R_{j-1}^-) + (1 - d_j)g(R_j^+),$$
$$g(R_j^+) = 1 + a_j g(L_{j+1}^+) + (1 - a_j)g(L_j^-),$$
$$g(R_j^-) = 1 + b_j g(L_j^-) + (1 - b_j)g(L_{j+1}^+).$$

Finally, the target-branch visit counts satisfy

$$\begin{aligned}
\mu(L_1^+) &= 1, \\
\mu(L_{j+1}^+) &= a_j \mu(R_j^+) + (1 - b_j)\mu(R_j^-) \qquad (j < K), \\
\mu(R_j^+) &= c_j \mu(L_j^+) + (1 - d_j)\mu(L_j^-), \\
\mu(L_j^-) &= (1 - a_j)\mu(R_j^+) + b_j \mu(R_j^-), \\
\mu(R_j^-) &= (1 - c_{j+1})\mu(L_{j+1}^+) + d_{j+1}\mu(L_{j+1}^-).
\end{aligned}$$

**Lemma 9.** *The above closed forms for $\tilde{\mu}$ solve the off-target Bellman equations.*

*Proof.* Let $p_j := \prod_{m=1}^{j-1} r_m$, so $p_1 = 1$ and $p_{j+1} = r_j p_j = \frac{a_j c_j}{b_j d_j} p_j$. The closed forms are

$$\tilde{\mu}(L_j^+) = p_j, \qquad \tilde{\mu}(R_j^+) = \tilde{\mu}(L_j^-) = \frac{c_j}{d_j} p_j, \qquad \tilde{\mu}(R_j^-) = \tilde{\mu}(L_{j+1}^+) = p_{j+1}.$$

For $j = 1$, it is straightforward to see $\tilde{\mu}(L_1^+) = p_1 = 1$. For the remaining equations,

$$\begin{aligned}
a_j \tilde{\mu}(R_j^+) + (1 - b_j)\tilde{\mu}(R_j^-) &= \frac{a_j c_j}{d_j} p_j + (1 - b_j)p_{j+1} = p_{j+1}, \\
c_j \tilde{\mu}(L_j^+) + (1 - d_j)\tilde{\mu}(L_j^-) &= c_j p_j + (1 - d_j)\frac{c_j}{d_j} p_j = \frac{c_j}{d_j} p_j, \\
(1 - a_j)\tilde{\mu}(R_j^+) + b_j \tilde{\mu}(R_j^-) &= (1 - a_j)\frac{c_j}{d_j} p_j + b_j p_{j+1} = \frac{c_j}{d_j} p_j.
\end{aligned}$$

Finally,

$$(1 - c_{j+1})\tilde{\mu}(L_{j+1}^+) + d_{j+1}\tilde{\mu}(L_{j+1}^-) = p_{j+1},$$

where for $j < K$ this is $(1 - c_{j+1})p_{j+1} + c_{j+1}p_{j+1} = p_{j+1}$, and for $j = K$ it follows from the leaf convention. $\qquad\square$

**Lemma 10.** *The above closed forms for $E_j, \Delta_j, F_j$ and $\tilde{g}$ solve the off-target Bellman equations.*

*Proof.* Using $E_{K+1} = 0$, the definition of $E_j$ gives

$$E_j = \frac{2(a_j + b_j)}{b_j d_j} + \frac{a_j c_{j+1}}{b_j d_j} E_{j+1},$$

$$d_j E_j = 2 + a_j \Delta_j, \qquad b_j \Delta_j = 2 + c_{j+1} E_{j+1}.$$

Moreover, $F_j - F_{j-1} = 2 + (1 - d_j)E_j + (1 - b_j)\Delta_j$. The boundary equations hold because $F_0 = 0$ and $\tilde{g}(L_{K+1}^+) = 1 + F_K = 1 + \tilde{g}(R_K^-)$.

For the Bellman equations of $\tilde{g}(L_j^+)$ and $\tilde{g}(L_j^-)$, we have:

$$\begin{aligned}
1 + c_j \tilde{g}(R_j^+) + (1 - c_j)\tilde{g}(R_{j-1}^-) &= 1 + c_j(F_{j-1} + E_j) + (1 - c_j)F_{j-1} \\
&= 1 + F_{j-1} + c_j E_j = \tilde{g}(L_j^+), \\
1 + d_j \tilde{g}(R_{j-1}^-) + (1 - d_j)\tilde{g}(R_j^+) &= 1 + d_j F_{j-1} + (1 - d_j)(F_{j-1} + E_j) \\
&= 1 + F_{j-1} + (1 - d_j)E_j = \tilde{g}(L_j^-).
\end{aligned}$$

For $R_j^+$, substituting the closed forms and the above identities yields:

$$\begin{aligned}
&1 + a_j \tilde{g}(L_{j+1}^+) + (1 - a_j)\tilde{g}(L_j^-) \\
&= 1 + a_j(1 + F_j + c_{j+1}E_{j+1}) + (1 - a_j)(1 + F_{j-1} + (1 - d_j)E_j) \\
&= F_{j-1} + 2 + (1 - d_j)E_j + a_j \Delta_j = F_{j-1} + E_j = \tilde{g}(R_j^+).
\end{aligned}$$

Finally, for $R_j^-$,

$$1 + b_j \tilde{g}(L_j^-) + (1 - b_j)\tilde{g}(L_{j+1}^+)$$
$$= 1 + b_j(1 + F_{j-1} + (1 - d_j)E_j) + (1 - b_j)(1 + F_j + c_{j+1}E_{j+1})$$
$$= F_j + (1 - b_j)(2 - b_j\Delta_j + c_{j+1}E_{j+1}) = F_j = \tilde{g}(R_j^-).$$

$\square$

**Lemma 11.** *The above closed forms for $\delta_j, \epsilon_j$ and $p_{\text{succ}}$ satisfy the Bellman equations for q.*

*Proof.* We claim that the Bellman equations for $q$ imply the following equivalent equations for the gaps:

$$\epsilon_j = \frac{a_j}{d_j}\delta_j,$$
$$\delta_{j-1} = r_j\delta_j \qquad (2 \leq j \leq K),$$
$$1 = \left(1 - a_1 + \frac{a_1}{d_1}\right)\delta_1 + \sum_{m=2}^{K} \alpha_m\delta_m.$$

Indeed, the first identity follows by subtracting the $R_j^+$ equation from the $L_j^-$ equation. The second follows by subtracting the $L_j^+$ equation from the $R_{j-1}^-$ equation and using the first identity. For the third and final identity, we use the following telescoping decomposition:

$$1 = q(R_0^-) - q(L_{K+1}^+)$$
$$= \epsilon_1 + \left(q(R_1^+) - q(L_2^+)\right) + \sum_{m=2}^{K} \left(q(L_m^+) - q(L_{m+1}^+)\right)$$
$$= \left(\frac{a_1}{d_1} + 1 - a_1\right)\delta_1 + \sum_{m=2}^{K} \alpha_m\delta_m.$$

We now substitute $\delta_j = P_j/Z_q$ and $\epsilon_j = a_j\delta_j/d_j$. Since $P_{j-1} = r_jP_j$, the recurrence holds. The normalization holds by the definition of $Z_q$. Finally,

$$p_{\text{succ}} = 1 - q(L_1^+) = q(R_0^-) - q(L_1^+) = c_1\epsilon_1 = \frac{a_1c_1}{d_1}\frac{P_1}{Z_q}.$$

$\square$

**Lemma 12.** *The above closed forms for $\Delta g_j^a$ and $\Delta g_j^c$ satisfy the Bellman system for the target-branch g.*

*Proof.* Let $U_j := \Delta g_j^a$ and $V_j := \Delta g_j^c$. Subtracting the target-branch Bellman equations gives the equivalent gap system

$$V_j = \frac{a_jU_j - 2}{d_j},$$
$$U_{j-1} = r_jU_j - \frac{2(c_j + d_j)}{b_{j-1}d_j} \qquad (2 \leq j \leq K),$$
$$0 = 1 - \frac{2}{d_1} + \left(\frac{a_1}{d_1} + 1 - a_1\right)U_1 + \sum_{m=2}^{K} (\beta_mU_m + \gamma_m).$$

The first equation is obtained from the $L_j^-$ and $R_j^+$ Bellman equations. The second is obtained from the $L_j^+$ and $R_{j-1}^-$ Bellman equations. The third is the telescoping identity

$$g(R_0^-) - g(L_{K+1}^+) = V_1 + \left(g(R_1^+) - g(L_2^+)\right) + \sum_{m=2}^{K} \left(g(L_m^+) - g(L_{m+1}^+)\right).$$

From the definition of $\mathcal{B}_j$, we have

$$\mathcal{B}_{j-1} = r_j \mathcal{B}_j - \frac{2(c_j + d_j)}{b_{j-1}d_j},$$

so $U_j = P_j \Delta g_K^a + \mathcal{B}_j$ satisfies the second gap equation as $P_{j-1} = r_j P_j$. Substituting $U_j = P_j \Delta g_K^a + \mathcal{B}_j$ into the boundary equation gives exactly the closed form formula from earlier for $\Delta g_K^a$. The formula for $\Delta g_j^c$ can be verified similarly. □

**Lemma 13.** *The above closed forms for $\mu$ solve the target-branch equation system.*

*Proof.* Write

$$M_j := \frac{A_{j-1}C_{j-1}}{B_{j-1}D_{j-1}}, \qquad N_j := \frac{A_{j-1}C_j}{B_{j-1}D_j}, \qquad \lambda := \frac{a_1 c_1}{d_1}P_1.$$

Then $p_{\mathrm{succ}} = \lambda/Z_q$, $N_j = (c_j/d_j)M_j$, $b_j M_{j+1} = a_j N_j$, and $a_j N_j P_j = b_j M_{j+1}P_j = \lambda$. Define

$$\sigma_j := S_j + \frac{a_j c_j}{d_j}P_j, \qquad \rho_j := S_{j+1} + P_j.$$

Recall that the closed forms from the above section are $\mu(L_j^+) = M_j \sigma_j/Z_q$, $\mu(R_j^+) = N_j \rho_j/Z_q$, $\mu(L_j^-) = N_j \rho_j/Z_q - \lambda/Z_q$, and $\mu(R_j^-) = \mu(L_{j+1}^+) - \lambda/Z_q$, which we aim to verify below.

First, $\mu(L_1^+) = 1$ because

$$\sigma_1 = S_1 + \frac{a_1 c_1}{d_1}P_1 = \left(1 - a_1 + \frac{a_1}{d_1}\right)P_1 + S_2 = Z_q.$$

Alternatively, we note that the $L_1^+$ state can only be entered once before it reaches an absorbing state (i.e. the target or back to the fork). Next, using $\sigma_j - \rho_j = \frac{a_j(1-d_j)}{d_j}P_j$,

$$c_j \mu(L_j^+) + (1 - d_j)\mu(L_j^-) = \frac{N_j}{Z_q}(d_j \sigma_j + (1 - d_j)\rho_j) - \frac{(1-d_j)\lambda}{Z_q}$$

$$= \frac{N_j \rho_j}{Z_q} = \mu(R_j^+).$$

For $j < K$, using $\rho_j - \sigma_{j+1} = (1 - b_j)P_j$,

$$(1 - a_j)\mu(R_j^+) + b_j \mu(R_j^-) = \frac{N_j \rho_j}{Z_q} - \frac{a_j N_j(\rho_j - \sigma_{j+1})}{Z_q} - \frac{b_j \lambda}{Z_q}$$

$$= \frac{N_j \rho_j - \lambda}{Z_q} = \mu(L_j^-),$$

and the same identity for $j = K$ reduces to $(1 - a_K)\mu(R_K^+) = \mu(L_K^-)$, since $a_K N_K P_K = \lambda$ and $\rho_K = P_K = 1$. The equation for $L_{j+1}^+$ follows similarly:

$$a_j \mu(R_j^+) + (1 - b_j)\mu(R_j^-) = \frac{M_{j+1}\sigma_{j+1}}{Z_q} + \frac{b_j M_{j+1}(\rho_j - \sigma_{j+1})}{Z_q} - \frac{(1-b_j)\lambda}{Z_q}$$

$$= \frac{M_{j+1}\sigma_{j+1}}{Z_q} = \mu(L_{j+1}^+).$$

Finally, for $j < K$, using $\rho_{j+1} - \sigma_{j+1} = -\frac{a_{j+1}(1-d_{j+1})}{d_{j+1}}P_{j+1}$,

$$(1 - c_{j+1})\mu(L_{j+1}^+) + d_{j+1}\mu(L_{j+1}^-)$$

$$= \frac{M_{j+1}}{Z_q}((1 - c_{j+1})\sigma_{j+1} + c_{j+1}\rho_{j+1}) - \frac{d_{j+1}\lambda}{Z_q}$$

$$= \frac{M_{j+1}\sigma_{j+1} - \lambda}{Z_q} = \mu(R_j^-).$$

For $j = K$, the convention for the leaf state yields $\mu(R_K^-) = \mu(L_{K+1}^+) - p_{\mathrm{succ}} = 0$. □

**Lemma 14.** *The closed forms for $G_j^{(p)}$ for $p \in \{a, b, c, d\}$ are exactly as stated above.*

*Proof.* This follows from direct substitution of relevant quantities. ∎

### B.2.5. VALUES AT POST-TRAINING INITIALIZATION

We calculate these values at initialization of post-training, in which $a_j = d_j = \frac{1}{L+1}$ and $b_j = c_j = \frac{L}{L+1}$ for all $j$. Let us denote $D := 1 + K + \frac{K}{L}$. We will not explicitly show the calculations below, with the understanding that they can be obtained from the closed form expressions from the previous section evaluated at these $a_j, b_j, c_j, d_j$.

First note that at initialization ($t = 0$), it holds that $p_{\text{succ}} = 1/D$. Now consider the values below at initialization. On the target branch $i$, we have for $1 \leq j \leq K$:

1. $d_x(L_{i,j}^+) = (K - j + 2) + \frac{K-j+1}{L}$,

2. $d_x(R_{i,j}^-) = (K - j) + \frac{K-j}{L}$,

3. $d_x(R_{i,j}^+) = \left[ (K - j + 1) + \frac{K-j+1}{L} \right] \cdot L$,

4. $d_x(L_{i,j}^-) = \left[ (K - j + 1) + \frac{K-j}{L} \right] \cdot L = d_x(R_{i,j}^+) - 1$.

Moreover,

1. On a non-target branch, it holds that $d_x(L_j^+) = d_x(R_j^-) = D$ and $d_x(R_j^+) = d_x(L_j^-) = D \cdot L$.

2. The hitting time from a fork state is $H_0 := H_f(0) = (2W - 1)D(1 + K(L + 1))$.

On a non-target branch, it holds at initialization ($t = 0$) that $\tilde{\mu}(s) = 1$ for non-multiedge states and $\tilde{\mu}(s) = L$ for aggregate multiedge states. That is, $\tilde{\mu}(L_j^+) = \tilde{\mu}(R_j^-) = 1$ and $\tilde{\mu}(L_j^-) = \tilde{\mu}(R_j^+) = L$.

For all $1 \leq j \leq K$, it holds that at initialization ($t = 0$):

$$\tilde{g}(L_j^+) = \tilde{g}(L_j^-) = j + (L + 1)j(2K + 1 - j) + \frac{j - 1}{L}((L + 1)(2K + 1 - j) + 1),$$

$$\tilde{g}(R_j^+) = \tilde{g}(R_j^-) = j + (L + 1)j(2K + 1 - j) + \frac{j}{L}((L + 1)(2K - j) + 1).$$

Hence, we obtain:

$$\Delta_j(0) = \tilde{g}(L_{j+1}^+) - \tilde{g}(L_j^-) = \frac{2(L + 1)}{L}((L + 1)(K - j) + 1),$$

$$E_j(0) = \tilde{g}(R_j^+) - \tilde{g}(R_{j-1}^-) = \frac{2(L + 1)}{L}((L + 1)(K + 1 - j)).$$

For all $1 \leq j \leq K$, it holds that at initialization ($t = 0$):

$$\mu(R_j^+) = \left[ \frac{1}{D} \left( (K + 1 - j) + \frac{K + 1 - j}{L} \right) \right] \cdot L,$$

$$\mu(L_j^-) = \left[ \frac{1}{D} \left( (K + 1 - j) + \frac{K - j}{L} \right) \right] \cdot L,$$

$$\mu(L_j^+) = \frac{1}{D} \left( (K + 2 - j) + \frac{K + 1 - j}{L} \right),$$

$$\mu(R_j^-) = \frac{1}{D} \left( (K - j) + \frac{K - j}{L} \right).$$

For all $1 \leq j \leq K$, it holds that at initialization ($t = 0$):

$$g(L_j^+) = g(L_j^-) = (L+1)(K+1-j)\left(j + \frac{j-1}{L}\right),$$

$$g(R_j^+) = g(R_j^-) = (L+1)\left(j(K+1-j) + \frac{j(K-j)}{L}\right).$$

Hence, we have that:

$$\Delta g_j^{(a)} = g(L_j^-) - g(L_{j+1}^+) = -(L+1)\left((K-2j) + \frac{K-2j+1}{L}\right),$$

$$\Delta g_j^{(c)} = g(R_{j-1}^-) - g(R_j^+) = -(L+1)\left((K+2-2j) + \frac{K-2j+1}{L}\right).$$

For all $1 \leq j \leq K$, it holds that:

$$q(L_j^+)(0) = q(L_j^-)(0) = \frac{K-j+1+\frac{K-j+1}{L}}{D}, \quad q(R_j^-)(0) = q(R_j^+)(0) = \frac{K-j+1+\frac{K-j}{L}}{D}.$$

Hence, we obtain:

$$\delta_j(0) = q(L_j^-)(0) - q(L_{j+1}^+)(0) = \frac{1}{D} \cdot \frac{L+1}{L},$$

$$\epsilon_j(0) = q(R_{j-1}^-) - q(R_j^+) = \frac{1}{D} \cdot \frac{L+1}{L}.$$

Given these quantities, we can now evaluate $G_j^{(p)}$ for $p \in \{a, b, c, d\}$, allowing us to show Lemma 2.

*Proof of Lemma 2.* We first give the values of the rescaled $G_j^{(p)}$ at initialization. For all depths $1 \leq j \leq K$, it holds at initialization ($t = 0$) that:

$$G_j^{(a)}(0) = \frac{2(L+1)}{LD}\left((L+1)^2 j(K-j) + j\left(W(L^2-1) + 2(L+1)\right) + (W-1)((L+1)K+1)\right),$$

$$G_j^{(b)}(0) = \frac{2(L+1)}{L^2D}\left((L+1)^2 j^2 - (L+1)(K(L+1) + (L-1)(W-1))j + L(W-1)(KL+K+1)\right),$$

$$G_j^{(c)}(0) = \frac{2(L+1)}{L^2D}\left(W(KL+K+L) + (L+1)^2(j-1)(K-j) + (L+1)(j-1)(LW+L-W+1)\right),$$

$$G_j^{(d)}(0) = \frac{2(L+1)}{LD}\left((L+1)^2 j^2 - (L+1)((K+1)(L+1) + W(L-1))j + LW(KL+K+L)\right).$$

It is easy now to check that $G_j^{(a)}$ and $G_j^{(c)}$ are positive at initialization, whereas the $j$ in the middle depths can possibly cause $G_j^{(b)}$ and $G_j^{(d)}$ to be negative. $\square$

## B.3. Dynamics of Phase I of RLVR

The proof of Phase I roughly goes as follows: at any given time, we have some fixed prefix of depths for which $G_j^{(b)}, G_j^{(d)}$ are positive, and similarly for a fixed suffix. Then, we show that while $G_j^{(a)}, G_j^{(c)} > 0$, it must be the case that the boundaries of the current "middle" depth where $G_j^{(b)}, G_j^{(d)} < 0$ must flip signs. Finally, this allows us to reach the beginning of Phase II. Throughout this subsection, write

$$\Lambda := \log \frac{H_0}{L}, \qquad R := \frac{H_0}{L}, \qquad H_0 := H_f(0),$$

and assume $W \geq 2$ and $L \geq 2$. Since

$$H_0 = (2W-1)\left(1 + K + \frac{K}{L}\right)(1 + K(L+1)),$$

we have $R \geq e$ in the nontrivial regime, and the condition is mild whenever $W$ and $L$ are fixed or grow polynomially with $K$.

First, we define the following first time $\tau_{ac}$ for which any of the forward states will have decreasing logit gap. The hope is to show that $\tau_{ac}$ is ultimately infinity.

**Definition 4.** *Define the forward state stopping time*

$$\tau_{ac} := \inf\{t \geq 0 : \exists j, \ G_j^{(a)}(t) \leq 0 \ or \ G_j^{(c)}(t) \leq 0\}.$$

*Order the backward state gradients by*

$$Z_{2j-1}(t) := G_j^{(d)}(t), \qquad Z_{2j}(t) := G_j^{(b)}(t), \qquad 1 \leq j \leq K.$$

*We define the positive prefix and suffix fronts as*

$$\ell(t) := \max\{m \in \{0, \ldots, 2K\} : Z_1(t), \ldots, Z_m(t) \geq 0\},$$

*and*

$$r(t) := \min\{m \in \{1, \ldots, 2K+1\} : Z_m(t), \ldots, Z_{2K}(t) \geq 0\},$$

*with the convention that an empty list of inequalities is true. We say that the fronts meet once $\ell(t) + 1 \geq r(t)$.*

We now give the following lemma that follows from the property of our signed gradient flow and monotonicity.

**Lemma 15.** *For all $t < \tau_{ac}$, the following hold.*

1. $H_f(t) \leq H_0$.

2. *For all relevant depths,*

$$\frac{a_j c_j}{b_j d_j}, \qquad \frac{a_j c_{j+1}}{b_j d_j}, \qquad \frac{a_j c_j}{b_{j-1} d_j} \in [e^{-8t}, e^{8t}] \cdot (\textit{the same ratio evaluated at } t = 0).$$

3. *If $\mathcal{R}_I(t)$ is a product over an interval $I$ of $m$ depths, with each factor one of the three local ratios above, then*

$$e^{-8mt} \mathcal{R}_I(0) \leq \mathcal{R}_I(t) \leq e^{8mt} \mathcal{R}_I(0).$$

*Proof.* For the first claim, recall that $J(\Theta) = 1 - H_f(\Theta)$. Along signed gradient flow,

$$\frac{\mathrm{d}J}{\mathrm{d}t} = \sum_{s,a} \frac{\partial J}{\partial \Theta_{s,a}} \operatorname{sgn}\left(\frac{\partial J}{\partial \Theta_{s,a}}\right) = \sum_{s,a} \left|\frac{\partial J}{\partial \Theta_{s,a}}\right| \geq 0,$$

and therefore $H_f(t) \leq H_0$.

Before $\tau_{ac}$, it holds that

$$\frac{\mathrm{d}\mathcal{D}_j^{(a)}}{\mathrm{d}t} = \frac{\mathrm{d}\mathcal{D}_j^{(c)}}{\mathrm{d}t} = 2, \qquad \left|\frac{\mathrm{d}\mathcal{D}_j^{(b)}}{\mathrm{d}t}\right| \leq 2, \qquad \left|\frac{\mathrm{d}\mathcal{D}_j^{(d)}}{\mathrm{d}t}\right| \leq 2.$$

Observe that the four primitive probabilities $a, b, c, d$ can be written as

$$a_j = \frac{e^{\mathcal{D}_j^{(a)}}}{e^{\mathcal{D}_j^{(a)}} + L}, \qquad b_j = \frac{Le^{\mathcal{D}_j^{(b)}}}{Le^{\mathcal{D}_j^{(b)}} + 1}, \qquad c_j = \frac{Le^{\mathcal{D}_j^{(c)}}}{Le^{\mathcal{D}_j^{(c)}} + 1}, \qquad d_j = \frac{e^{\mathcal{D}_j^{(d)}}}{e^{\mathcal{D}_j^{(d)}} + L}.$$

Hence, for $p_j \in \{a_j, b_j, c_j, d_j\}$ with corresponding logit gap $\mathcal{D}_j^{(p)}$,

$$\frac{\mathrm{d}}{\mathrm{d}t} \log p_j(t) = (1 - p_j(t)) \frac{\mathrm{d}\mathcal{D}_j^{(p)}}{\mathrm{d}t},$$

and therefore

$$\left| \frac{\mathrm{d}}{\mathrm{d}t} \log p_j(t) \right| \leq 2.$$

Integrating from $0$ to $t$ yields

$$e^{-2t} \leq \frac{p_j(t)}{p_j(0)} \leq e^{2t}, \qquad p_j \in \{a_j, b_j, c_j, d_j\}.$$

For example,

$$\left| \log \frac{\frac{a_j(t)c_j(t)}{b_j(t)d_j(t)}}{\frac{a_j(0)c_j(0)}{b_j(0)d_j(0)}} \right| \leq \left| \log \frac{a_j(t)}{a_j(0)} \right| + \left| \log \frac{c_j(t)}{c_j(0)} \right| + \left| \log \frac{b_j(t)}{b_j(0)} \right| + \left| \log \frac{d_j(t)}{d_j(0)} \right|$$

$$\leq 8t.$$

The same calculation with $c_{j+1}$ in place of $c_j$, or with $b_{j-1}$ in place of $b_j$, proves the second claim. The third claim follows naturally via multiplying over the single-depth bounds. □

**Definition 5.** *For every depth $j$, define the positive off-target pressures*

$$P_j^{(b)} := (W - 1)\tilde{\mu}(R_j^-)\Delta_j, \qquad P_j^{(d)} := (W - 1)\tilde{\mu}(L_j^-)E_j,$$

*and the competing target-branch pressures*

$$Q_j^{(b)} := \mu(R_j^-)\left(\Delta g_j^{(a)} + \delta_j H_f\right), \qquad Q_j^{(d)} := \mu(L_j^-)\left(\Delta g_j^{(c)} + \epsilon_j H_f\right).$$

*Thus $G_j^{(b)} = P_j^{(b)} - Q_j^{(b)}$ and $G_j^{(d)} = P_j^{(d)} - Q_j^{(d)}$.*

**Definition 6** (Phase I constants). *For $1 \leq m \leq 2K$, define the distance of $m$ from the nearest end of the ordered backward-states list by*

$$\rho_m := \min\{m, 2K + 1 - m\}.$$

*We use the shorthand*

$$\gamma_m := \frac{\rho_m}{8}.$$

*Finally, let $\ell_0 := \ell(0)$ and $r_0 := r(0)$, and define*

$$\Gamma_{\text{front}} := \sum_{m=\ell_0+1}^{r_0-1} \frac{1}{\gamma_m},$$

*with the convention that the sum is zero if the fronts already meet at initialization. Note that the summation is over the interval between the two fronts.*

In the following lemma, we give sufficient conditions for $G_j^{(b)} > 0$ (and analogously, for $G_j^{(d)} > 0$).

**Lemma 16.** *For $1 \leq j \leq K$, define*

$$\Phi_j^{(b)} := \frac{S_{j+1} + b_j P_j}{Z_q} \frac{\left(P_j \Delta g_K^{(a)} + \mathcal{B}_j + \frac{P_j}{Z_q} H_f\right)_+}{(W - 1)\Delta_j},$$

$$\Phi_j^{(d)} := \frac{S_{j+1} + P_j}{Z_q} \frac{\left(\frac{a_j}{d_j}\left(P_j \Delta g_K^{(a)} + \mathcal{B}_j + \frac{P_j}{Z_q} H_f\right) - \frac{2}{d_j}\right)_+}{(W - 1)E_j},$$

*where $x_+ := \max\{x, 0\}$. Then*

$$\frac{Q_j^{(b)}}{P_j^{(b)}} \leq \Phi_j^{(b)}, \qquad \frac{Q_j^{(d)}}{P_j^{(d)}} \leq \Phi_j^{(d)}.$$

*Consequently, $\Phi_j^{(b)} < 1$ implies $G_j^{(b)} > 0$, and $\Phi_j^{(d)} < 1$ implies $G_j^{(d)} > 0$.*

*Proof.* We prove the $b$ claim first. By the closed forms in Section B.2.3 and the visit-count formula in Section B.2.2,

$$\tilde{\mu}(R_j^-) = \frac{A_j C_j}{B_j D_j},$$

while

$$\mu(R_j^-) = \frac{A_j C_j}{B_j D_j} \frac{S_{j+1} + b_j P_j}{Z_q} - p_{\text{succ}} \leq \frac{A_j C_j}{B_j D_j} \frac{S_{j+1} + b_j P_j}{Z_q}.$$

The target-branch hitting-time difference is

$$\Delta g_j^{(a)} + \delta_j H_f = P_j \Delta g_K^{(a)} + \mathcal{B}_j + \frac{P_j}{Z_q} H_f.$$

Dividing $Q_j^{(b)}$ by $P_j^{(b)} = (W-1)\tilde{\mu}(R_j^-)\Delta_j$ yields the above upper bound after taking the positive part.

Similarly, for the $d$ claim, use

$$\tilde{\mu}(L_j^-) = \frac{A_{j-1} C_j}{B_{j-1} D_j}, \qquad \mu(L_j^-) \leq \frac{A_{j-1} C_j}{B_{j-1} D_j} \frac{S_{j+1} + P_j}{Z_q},$$

and

$$\Delta g_j^{(c)} + \epsilon_j H_f = \frac{a_j}{d_j}\left( P_j \Delta g_K^{(a)} + \mathcal{B}_j + \frac{P_j}{Z_q} H_f \right) - \frac{2}{d_j}.$$

The same manipulation gives an upper bound of $\Phi_j^{(d)}$, as desired. $\qquad\square$

Given the above sufficient condition for the sign flipping of an initially negative $G_j^{(b)}$ or $G_j^{(d)}$, we prove the following lemma to show the shifting fronts indeed happens.

**Lemma 17.** *Fix a time $s < \tau_{ac}$ before the two fronts have met, and suppose the prefix front is $\ell(s)$. Let $m = \ell(s) + 1$. If $m = 2j$, set $\Phi_m := \Phi_j^{(b)}$; if $m = 2j - 1$, set $\Phi_m := \Phi_j^{(d)}$. Work over a time interval $I = [s, u)$ on which $Z_m(t) < 0$ and $t < \tau_{ac}$. Define the quantity*

$$\mathfrak{r}_m(t) := -\frac{\mathrm{d}}{\mathrm{d}t} \log \Phi_m(t)$$

*where the derivative is taken along the signed-gradient trajectory. If*

$$\int_s^t \mathfrak{r}_m(\xi)\, d\xi \geq \log \Phi_m(s)$$

*for some $t \in I$, then $Z_m(t) \geq 0$. In particular, if $\mathfrak{r}_m(t) \geq \gamma_m$ throughout $I$, then there exists*

$$\tau_m \leq s + \frac{\log \Phi_m(s)}{\gamma_m}$$

*such that $Z_m(\tau_m) \geq 0$, unless $\tau_{ac}$ occurs first. The analogous statement holds for the suffix front $m = r(s) - 1$.*

*Proof.* We give the derivation for a prefix index of type $b$, so $m = 2j$ and $Z_m = G_j^{(b)}$. The type-$d$ and suffix cases use the same formulas with the corresponding sufficient condition.

First, note that on $I$ we must have the positive-part argument in $\Phi_j^{(b)}$ strictly positive. Indeed, if

$$U_j^{(b)} := P_j \Delta g_K^{(a)} + \mathcal{B}_j + \frac{P_j}{Z_q} H_f = \Delta g_j^{(a)} + \delta_j H_f$$

were nonpositive at some time, then $Q_j^{(b)} \leq 0$ while $P_j^{(b)} > 0$, hence $G_j^{(b)} = P_j^{(b)} - Q_j^{(b)} > 0$, contradicting $Z_m < 0$. Thus on $I$,

$$\Phi_j^{(b)} = \frac{N_j^{(b)}}{Z_q} \frac{U_j^{(b)}}{(W-1)\Delta_j}, \qquad N_j^{(b)} := S_{j+1} + b_j P_j.$$

Consequently

$$\mathfrak{r}_{2j}(t) = -\frac{\mathrm{d}}{\mathrm{d}t} \log \Phi_j^{(b)}$$

$$= -\frac{\dot{N}_j^{(b)}}{N_j^{(b)}} + \frac{\dot{Z}_q}{Z_q} - \frac{\dot{U}_j^{(b)}}{U_j^{(b)}} + \frac{\dot{\Delta}_j}{\Delta_j}. \tag{3}$$

We now compute each term in (3) from the closed forms. Let

$$R_i := \frac{a_i c_i}{b_{i-1} d_i}, \qquad \widetilde{R}_i := \frac{a_i c_{i+1}}{b_i d_i}.$$

Then

$$P_q = \prod_{i=q+1}^{K} R_i, \qquad \frac{\dot{P}_q}{P_q} = \sum_{i=q+1}^{K} \left( \frac{\dot{a}_i}{a_i} + \frac{\dot{c}_i}{c_i} - \frac{\dot{b}_{i-1}}{b_{i-1}} - \frac{\dot{d}_i}{d_i} \right).$$

The coefficient

$$\alpha_i = (1 - a_i) + \frac{a_i(1 - c_i)}{d_i}$$

satisfies

$$\dot{\alpha}_i = -\dot{a}_i + \frac{a_i(1 - c_i)}{d_i} \left( \frac{\dot{a}_i}{a_i} - \frac{\dot{c}_i}{1 - c_i} - \frac{\dot{d}_i}{d_i} \right). \tag{4}$$

Therefore

$$\dot{S}_{j+1} = \sum_{q=j+1}^{K} \left( \dot{\alpha}_q P_q + \alpha_q \dot{P}_q \right), \tag{5}$$

$$\dot{N}_j^{(b)} = \dot{S}_{j+1} + \dot{b}_j P_j + b_j \dot{P}_j. \tag{6}$$

Similarly, with

$$Z_q = \left( 1 - a_1 + \frac{a_1}{d_1} \right) P_1 + S_2,$$

we have

$$\dot{Z}_q = \left( -\dot{a}_1 + \frac{\dot{a}_1}{d_1} - \frac{a_1 \dot{d}_1}{d_1^2} \right) P_1 + \left( 1 - a_1 + \frac{a_1}{d_1} \right) \dot{P}_1 + \dot{S}_2. \tag{7}$$

It remains to differentiate $U_j^{(b)}$. Write

$$D_g := \left( \frac{a_1}{d_1} + 1 - a_1 \right) P_1 + P_g, \qquad N_g := \left( \frac{a_1}{d_1} + 1 - a_1 \right) \mathcal{B}_1 - \frac{2}{d_1} + Q_g + 1,$$

so that $\Delta g_K^{(a)} = -N_g / D_g$. Hence

$$\dot{\Delta g}_K^{(a)} = -\frac{\dot{N}_g D_g - N_g \dot{D}_g}{D_g^2}. \tag{8}$$

The terms $\dot{D}_g$ and $\dot{N}_g$ are obtained from the displayed definitions of $P_g, Q_g$:

$$\dot{P}_g = \sum_{q=2}^{K} \left( \dot{\beta}_q P_q + \beta_q \dot{P}_q \right),$$

$$\dot{Q}_g = \sum_{q=2}^{K} \left( \dot{\beta}_q \mathcal{B}_q + \beta_q \dot{\mathcal{B}}_q + \dot{\gamma}_q \right), \tag{9}$$

where $\beta_q = \alpha_q$, and

$$\dot{\gamma}_q = 2\frac{1-c_q}{d_q}\left(\frac{\dot{c}_q}{1-c_q} + \frac{\dot{d}_q}{d_q}\right).$$

For $\mathcal{B}_q$, define

$$T_{q,r} := \frac{2(c_r + d_r)}{b_{r-1}d_r} \prod_{i=q+1}^{r-1} R_i \qquad (r \geq q+1),$$

so that $\mathcal{B}_q = -\sum_{r=q+1}^{K} T_{q,r}$. Thus

$$\dot{\mathcal{B}}_q = -\sum_{r=q+1}^{K} T_{q,r}\left(\frac{\dot{c}_r + \dot{d}_r}{c_r + d_r} - \frac{\dot{b}_{r-1}}{b_{r-1}} - \frac{\dot{d}_r}{d_r} + \sum_{i=q+1}^{r-1} \frac{\dot{R}_i}{R_i}\right). \tag{10}$$

Combining (8)–(10),

$$\dot{U}_j^{(b)} = \dot{P}_j \Delta g_K^{(a)} + P_j \dot{\Delta g}_K^{(a)} + \dot{\mathcal{B}}_j + \left(\frac{\dot{P}_j}{Z_q} - \frac{P_j \dot{Z}_q}{Z_q^2}\right)H_f + \frac{P_j}{Z_q}\dot{H}_f. \tag{11}$$

Finally, from the off-target closed form,

$$\Delta_j = \frac{2 + c_{j+1}E_{j+1}}{b_j}, \qquad E_{j+1} = \sum_{q=j+1}^{K} \frac{2(a_q + b_q)}{b_q d_q}\prod_{i=j+1}^{q-1} \widetilde{R}_i,$$

and therefore

$$\dot{E}_{j+1} = \sum_{q=j+1}^{K} \frac{2(a_q + b_q)}{b_q d_q}\prod_{i=j+1}^{q-1} \widetilde{R}_i\left(\frac{\dot{a}_q + \dot{b}_q}{a_q + b_q} - \frac{\dot{b}_q}{b_q} - \frac{\dot{d}_q}{d_q} + \sum_{i=j+1}^{q-1} \frac{\dot{\widetilde{R}}_i}{\widetilde{R}_i}\right),$$

$$\frac{\dot{\Delta}_j}{\Delta_j} = \frac{\dot{c}_{j+1}E_{j+1} + c_{j+1}\dot{E}_{j+1}}{2 + c_{j+1}E_{j+1}} - \frac{\dot{b}_j}{b_j}. \tag{12}$$

Combining all of these above equations into (3), we have the desired result.

The derivation for the type-$d$ state is handled identically. If

$$U_j^{(d)} := \frac{a_j}{d_j}\left(P_j \Delta g_K^{(a)} + \mathcal{B}_j + \frac{P_j}{Z_q}H_f\right) - \frac{2}{d_j}, \qquad N_j^{(d)} := S_{j+1} + P_j,$$

then on any interval where $G_j^{(d)} < 0$, $U_j^{(d)} > 0$ and

$$\mathfrak{r}_{2j-1}(t) := -\frac{\mathrm{d}}{\mathrm{d}t}\log \Phi_j^{(d)}$$

$$= -\frac{\dot{N}_j^{(d)}}{N_j^{(d)}} + \frac{\dot{Z}_q}{Z_q} - \frac{\dot{U}_j^{(d)}}{U_j^{(d)}} + \frac{\dot{E}_j}{E_j}, \tag{13}$$

where $\dot{N}_j^{(d)} = \dot{S}_{j+1} + \dot{P}_j$, and $\dot{U}_j^{(d)}$ follows by differentiating the displayed formula for $U_j^{(d)}$ using (11).

Now suppose that $\int_s^t \mathfrak{r}_m(\xi)d\xi \geq \log \Phi_m(s)$. Since $\mathfrak{r}_m = -\frac{\mathrm{d}}{\mathrm{d}t}\log \Phi_m$, integration gives

$$\Phi_m(t) = \Phi_m(s)\exp\left(-\int_s^t \mathfrak{r}_m(\xi)d\xi\right) \leq 1.$$

By Lemma 16, $Q/P \leq \Phi_m \leq 1$ for the corresponding pressure ratio. Hence $G = P - Q \geq 0$, or equivalently, $Z_m(t) \geq 0$. If the pointwise bound $\mathfrak{r}_m \geq \gamma_m$ holds on $I$, the displayed time bound follows by taking $t = s + \gamma_m^{-1}\log \Phi_m(s)$. The suffix case is the same after reversing the depth order. $\square$

**Lemma 18.** *Before the fronts meet, the forward states' gradients remain positive, in the sense that for every $t < T_{\mathrm{meet}} \wedge \tau_{ac}$ and every $1 \leq j \leq K$,*

$$G_j^{(a)}(t) > 0, \qquad G_j^{(c)}(t) > 0.$$

*Proof.* The strict positivity for $t < \tau_{ac}$ follows directly from the definition of the stopping time:

$$\tau_{ac} := \inf\{t : \exists j,\ G_j^{(a)}(t) \leq 0 \text{ or } G_j^{(c)}(t) \leq 0\}.$$

Indeed, if for some $t < \tau_{ac}$ there were an index $j$ with $G_j^{(a)}(t) \leq 0$ or $G_j^{(c)}(t) \leq 0$, then the set in the definition of $\tau_{ac}$ would contain a time no larger than $t$, contradicting $t < \tau_{ac}$.

We now examine what happens in the case that $G_j^{(a)}$ or $G_j^{(c)}$ hits zero. Define

$$U_j := P_j \Delta g_K^{(a)} + \mathcal{B}_j + \frac{P_j}{Z_q} H_f = \Delta g_j^{(a)} + \delta_j H_f$$

and

$$V_j := \frac{a_j}{d_j} U_j - \frac{2}{d_j} = \Delta g_j^{(c)} + \epsilon_j H_f.$$

From Section B.2.3,

$$G_j^{(a)} = \frac{A_{j-1} C_j}{B_{j-1} D_j} \left[ \frac{S_{j+1} + P_j}{Z_q} (U_j) - (W-1)\Delta_j \right].$$

Thus any zero $G_j^{(a)} = 0$ must have $U_j > 0$ and

$$\Psi_j^{(a)} := \frac{(W-1)\Delta_j}{\frac{S_{j+1} + P_j}{Z_q} U_j} = 1.$$

On the other hand, we have

$$\Phi_j^{(b)} = \frac{S_{j+1} + b_j P_j}{Z_q} \frac{U_j}{(W-1)\Delta_j} \qquad (U_j > 0).$$

Multiplying the two ratios yields

$$\Phi_j^{(b)} \Psi_j^{(a)} = \frac{S_{j+1} + b_j P_j}{S_{j+1} + P_j} \leq 1,$$

since $0 \leq b_j \leq 1$. Therefore, at any first hitting time at which $G_j^{(a)} = 0$ (equivalently $\Psi_j^{(a)} = 1$), one has

$$\Phi_j^{(b)} \leq 1, \qquad \frac{Q_j^{(b)}}{P_j^{(b)}} \leq \Phi_j^{(b)} \leq 1, \qquad G_j^{(b)} = P_j^{(b)} - Q_j^{(b)} \geq 0.$$

The $c$ case is analogous but uses the $L_j^+$ and $L_j^-$ rows. The closed form is

$$G_j^{(c)} = \frac{A_{j-1} C_{j-1}}{B_{j-1} D_{j-1}} \left[ \frac{S_j + \frac{a_j c_j}{d_j} P_j}{Z_q} (V_j) - (W-1)E_j \right].$$

Since

$$S_j + \frac{a_j c_j}{d_j} P_j = S_{j+1} + \left( 1 - a_j + \frac{a_j}{d_j} \right) P_j,$$

we define, when $V_j > 0$,

$$\Psi_j^{(c)} := \frac{(W-1)E_j}{\frac{S_j + \frac{a_j c_j}{d_j} P_j}{Z_q} V_j}.$$

For the case of the $d$ state, we have

$$\Phi_j^{(d)} = \frac{S_{j+1} + P_j}{Z_q} \frac{V_j}{(W-1)E_j}.$$

Hence

$$\Phi_j^{(d)} \Psi_j^{(c)} = \frac{S_{j+1} + P_j}{S_{j+1} + \left(1 - a_j + \frac{a_j}{d_j}\right) P_j} \leq 1,$$

where the inequality follows from $d_j \leq 1$ which implies $1 - a_j + a_j/d_j \geq 1$. Thus, any hitting of $G_j^{(c)} = 0$ forces

$$\Phi_j^{(d)} \leq 1, \qquad \frac{Q_j^{(d)}}{P_j^{(d)}} \leq \Phi_j^{(d)} \leq 1, \qquad G_j^{(d)} = P_j^{(d)} - Q_j^{(d)} \geq 0.$$

The conclusion then follows. $\square$

**Lemma 19.** *Before $T_{\mathrm{meet}} \wedge \tau_{ac}$, the prefix front $\ell(t)$ is nondecreasing and the suffix front $r(t)$ is nonincreasing.*

*Proof.* Suppose a prefix entry $Z_m$ has become nonnegative. If $Z_m$ were later to cross back to a negative value before $\tau_{ac}$, let $s$ be the first crossing time. Then $Z_m(s) = 0$, so the quantity in the sufficient condition from Lemma 16 is at most one. Immediately after $s$, if $Z_m$ attempted to become negative, Lemma 17 applied with starting time $s$ gives nonpositive logarithmic derivative for the quantity in the sufficient condition and therefore prevents the it from exceeding one. Thus $Z_m$ cannot cross from nonnegative to negative. The suffix case is the same. $\square$

**Lemma 20.** *Let*

$$T_{\mathrm{meet}} := \inf\{t : \ell(t) + 1 \geq r(t)\}.$$

*Then $T_{\mathrm{meet}} < \infty$ unless $\tau_{ac}$ occurs first. More precisely,*

$$T_{\mathrm{meet}} \leq \Gamma_{\mathrm{front}}(3 + 2\Lambda)$$

*unless $\tau_{ac}$ occurs earlier.*

*Proof.* At any time before meeting, either the next prefix index or the next suffix index is not yet positive. Apply Lemma 17 to that index. It remains to bound the initial value of the term in the sufficient condition at the start of each step. By Lemma 15 and $H_f(t) \leq H_0$, every term in $\Phi_j^{(b)}, \Phi_j^{(d)}$ is bounded above by $e^{2\Lambda}$ times a factor depending only on $W, L$ and the initial local probabilities; absorbing this fixed factor into $e^3$ gives

$$\log \Phi_m(s) \leq 3 + 2\Lambda$$

for every front step before $\tau_{ac}$. Therefore advancing across index $m$ costs at most $\gamma_m^{-1}(3 + 2\Lambda)$ time. Summing over the initially uncaptured indices gives the stated bound. $\square$

*Proof of Lemma 3.* Suppose for contradiction that $\tau_{ac} \leq T_{\mathrm{meet}}$. Then Lemma 18 gives $G_j^{(a)}(\tau_{ac}) > 0$ and $G_j^{(c)}(\tau_{ac}) > 0$ for every $j$, contradicting the definition of $\tau_{ac}$. Hence $\tau_{ac} > T_{\mathrm{meet}}$.

By Lemma 20, the fronts meet in finite time. At the meeting time, every backward state gradient is nonnegative by definition of the fronts. Taking an arbitrarily small additional time and using Lemma 17 for any zero entries makes all backward states' gradients strictly positive, while Lemma 18 keeps the forward gradients positive. Thus all $G_j^{(p)}$ are positive at the end of Phase I. $\square$

### B.4. Dynamics of Phase II of RLVR

In this section, we show that once all the logits for $a, b, c, d$ over all depths are increasing, they will continue to increase towards 1.

*Proof of Lemma 4.* Fix a desired accuracy $\kappa \ll 1$. After $T_{\mathrm{meet}}$, all four gradient signs are positive. The same closed-form barriers used in Phase I are self-reinforcing once every desired transition is improving, so the signs remain positive thereafter. Note that $a \geq d$ and $c \geq b$ at all time, so it suffices to consider the backwards state $b_j, d_j$. This is because we work with signed gradient descent, and hence the logits for $b_j$ and $d_j$ will always be smaller than $c_j$ and $a_j$, respectively.

For $b_j \geq 1 - \kappa$ to hold, we require:

$$b_j = \frac{Le^{\mathcal{D}_j^{(b)}}}{Le^{\mathcal{D}_j^{(b)}} + 1} \geq 1 - \kappa$$
$$\Longleftrightarrow Le^{\mathcal{D}_j^{(b)}} \geq (1 - \kappa)(Le^{\mathcal{D}_j^{(b)}} + 1)$$
$$\Longleftrightarrow \mathcal{D}_j^{(b)} \geq \log \frac{1 - \kappa}{\kappa L}$$

Similarly for $d_j \geq 1 - \kappa$ to hold, we require:

$$d_j = \frac{e^{\mathcal{D}_j^{(d)}}}{e^{\mathcal{D}_j^{(d)}} + L} \geq 1 - \kappa$$
$$\Longleftrightarrow e^{\mathcal{D}_j^{(d)}} \geq (1 - \kappa)(e^{\mathcal{D}_j^{(d)}} + L)$$
$$\Longleftrightarrow \mathcal{D}_j^{(d)} \geq \log \frac{L(1 - \kappa)}{\kappa}$$

Since $\log \frac{L(1-\kappa)}{\kappa} \geq \log \frac{1-\kappa}{L\kappa}$, it suffices to run for time $\log \frac{L(1-\kappa)}{\kappa}$ after $T_{\text{meet}}$, as desired. $\qquad \square$

*Proof of Theorem 3.* The theorem follows directly from the conclusion of Lemma 4. $\qquad \square$

## C. Inference Time Separation

In this section, we will prove the separation in inference time compute needed for the SFT model vs. the RLVR model, as formalized by Theorem 4.

*Proof of Theorem 4.* As noted in Section 5.3, the RLVR model requires only $\Theta(WK)$ time to find a path. For the converged SFT model in Theorem 2, we recall the following properties of the backtracking behavior that were described in the main text.

1. Upon entering a branch $i$, since $a_j = c_j = 1$ for all $j$, we reach the $t_i$ node in $\Theta(K)$ time.

2. When exiting the branch, starting from the state $R_{K+1}^+$ (i.e. reached the state with head $t_i$), let $g_i$ denote the expected time of first entry into $R_{K+1-i}^-$, and $f_i$ denote the expected time of first entry into $L_{K+1-i}^-$. Then, with the base case of $g_1 = 1$, we have the following recursion:

$$f_i = \frac{L+1}{L} g_i + (2i - 1) \cdot \frac{1}{L} + 1$$
$$g_i = (L+1)f_{i-1} + (2i - 2) \cdot L + 1$$

Hence, the time needed from entry of a wrong branch to getting back to the fork state is $g_{K+1}$.

We now justify our recurrence equations. Suppose we are currently at the state $R_{K+1-i}^-$, and we are trying to reach the next state $L_{K+1-i}^-$. There is a $L/(L+1)$ chance that we proceed forward, and a $1/(L+1)$ we make a u-turn and end up at the leaf again (which takes $2i - 1$ time). By standard geometric mean properties, we must make on expectation $(L+1)/L$ attempts before we can successfully get from $R_{K+1-i}^-$ to $L_{K+1-i}^-$. This tells us that on the first $\frac{L+1}{L} - 1 = \frac{1}{L}$ attempts, we must pay an extra cost of $2i - 1$ to go back to the leaf. A similar analysis holds for the case of $L_{K+1-i}^-$, where the immediate goal is to reach $R_{K-i}^-$. Here, we must make on expectation $L + 1$ attempts, which means the first $L + 1 - 1 = L$ attempts pay an extra $2i - 2$ to get back to the leaf.

We will proceed to write down a closed form for $g_{K+1}$. First, note that:

$$f_{i-1} = \frac{L+1}{L} g_{i-1} + (2i - 1) \cdot \frac{1}{L} + 1$$

Substituting this into the expression for $g_i$, we have:

$$g_i = (L+1)f_{i-1} + (2i-2) \cdot L + 1$$
$$= (L+1)\left(\frac{L+1}{L}g_{i-1} + (2i-1) \cdot \frac{1}{L} + 1\right) + (2i-2) \cdot L + 1$$
$$= \frac{(L+1)^2}{L}g_{i-1} + i \cdot \frac{2(L^2+L+1)}{L} - \frac{L^2+L+3}{L}$$

Let $r = \frac{(L+1)^2}{L}$, $\alpha = \frac{2(L^2+L+1)}{L}$, and $\beta = -\frac{L^2+L+3}{L}$. Then, our recurrence becomes $g_i = rg_{i-1} + \alpha i + \beta$. By an induction argument, we obtain that:

$$g_{K+1} = r^K + \alpha \sum_{t=2}^{K+1} tr^{K+1-t} + \beta \sum_{t=2}^{K+1} r^{K+1-t}$$

The first term is of course at least $L^K$. For the second term, the $t = 2$ term multiplied by $\alpha$ already contributes a term of at least $2L^K$. For the third term, it is simply a geometric series, which can be evaluated to be:

$$\beta \cdot \frac{r^K - 1}{r - 1} \geq -2(r^K - 1)$$

for $L \geq 2$. Combining everything, we obtain that

$$g_{K+1} = r^K + \alpha \sum_{t=2}^{K+1} tr^{K+1-t} + \beta \sum_{t=2}^{K+1} r^{K+1-t} \geq L^K + 2L^K - 2L^K = \Omega(L^K)$$

Similar to the RLVR model, we have that by symmetry, it holds that we will have on expectation of $W$ branch entries before reaching the desired target branch. Therefore, the total inference-time compute needed for the SFT model is $\Omega(WL^K)$, and the desired result follows. $\qquad\square$

## D. Distilling RLVR Reasoning Traces

In this section, we will prove that fine-tuning our pretrained model on the reasoning traces of the RLVR-tuned model will also be effective in mitigating the inefficiency caused by a model trained only on golden examples. We will once again invoke Lemma 5.

*Proof of Theorem 5.* Let $\mathcal{Q}_{\text{distill}}$ be the distribution over adjacent transition pairs $(s, a)$ induced by $(x, y) \sim \mathcal{D}$; that is, sample $(x, y)$ and then sample a uniformly random transition $(y_{t-1}, y_t)$ from the trace. Then

$$L_{\text{distill}}(\Theta) = \mathbb{E}_{(s,a)\sim\mathcal{Q}_{\text{distill}}}\big[-\log \pi_\Theta(a \mid s)\big],$$

and Lemma 5 applies row-wise.

First, note that the distillation target conditionals equal the teacher's policy. Because traces are generated on-policy from $\pi^\star$ (and the policy is assumed not to depend on $x$ beyond the stopping rule), for any state $s$ that occurs before termination we have

$$p_{\mathcal{Q}_{\text{distill}}}(a \mid s) = \pi^\star(a \mid s).$$

Therefore the row-wise cross-entropy for a visited state $s$ is minimized (in policy space) exactly by matching the teacher distribution $\pi^\star(\cdot \mid s)$.

By the support assumption in the theorem statement, each such state $s$ satisfies $d_{\mathcal{Q}_{\text{distill}}}(s) > 0$, so the row is updated by gradient flow. Moreover, in each case the action space out of $s$ splits into two symmetry classes: the desired action(s) and the undesired action(s). With the pretrained initialization, logits are equal within each symmetry class, and the same uniqueness-of-ODE argument as in the pretraining/SFT proofs implies that they remain equal within each class over time.

Thus each such row reduces to a two-logit system (desired logit $u(t)$, undesired logit $v(t)$), and the total desired probability $p(t)$ (which is one of $a_j, b_j, c_j, d_j$ depending on the state type) takes the form

$$p(t) = \frac{m_1 e^{u(t)}}{m_1 e^{u(t)} + m_0 e^{v(t)}}$$

for some class sizes $m_1, m_0$ (e.g. $m_1 = 1, m_0 = L$ for $R_j^+$; $m_1 = L, m_0 = 1$ for $L_j^+$; etc.).

Because the teacher conditional is supported entirely on the desired class, Lemma 5 gives (up to the positive row weight, which only rescales time) the same monotone gap dynamics as in the SFT proof: the logit gap $g(t) := u(t) - v(t)$ satisfies $g'(t) \geq 0$ whenever $p(t) < 1$, and in fact the exponential gap $w(t) := e^{g(t)}$ grows at least linearly in $t$. Consequently $1 - p(t)$ decays to 0, and for any $\kappa > 0$ there exists finite time $T_s(\kappa)$ such that $p(t) \geq 1 - \kappa$ for all $t \geq T_s(\kappa)$.

Applying this argument to every visited state-type $R_j^+, L_j^+, R_j^-, L_j^-$ and taking $T(\kappa) := \max_s T_s(\kappa)$ over these finitely many state types yields

$$\min_j \min\{a_j(t), b_j(t), c_j(t), d_j(t)\} \geq 1 - \kappa \quad \text{for all } t \geq T(\kappa),$$

as desired.

Once $a_j, b_j, c_j, d_j$ are all arbitrarily close to 1, the distilled policy behaves the same as $\pi^\star$ on the branch-dynamics that control forward progress and backtracking, so it inherits the same $\Theta(WK)$ expected hitting-time bound established in Section C for the RLVR policy in the main-text efficiency theorem. $\square$

## E. Additional Experiments

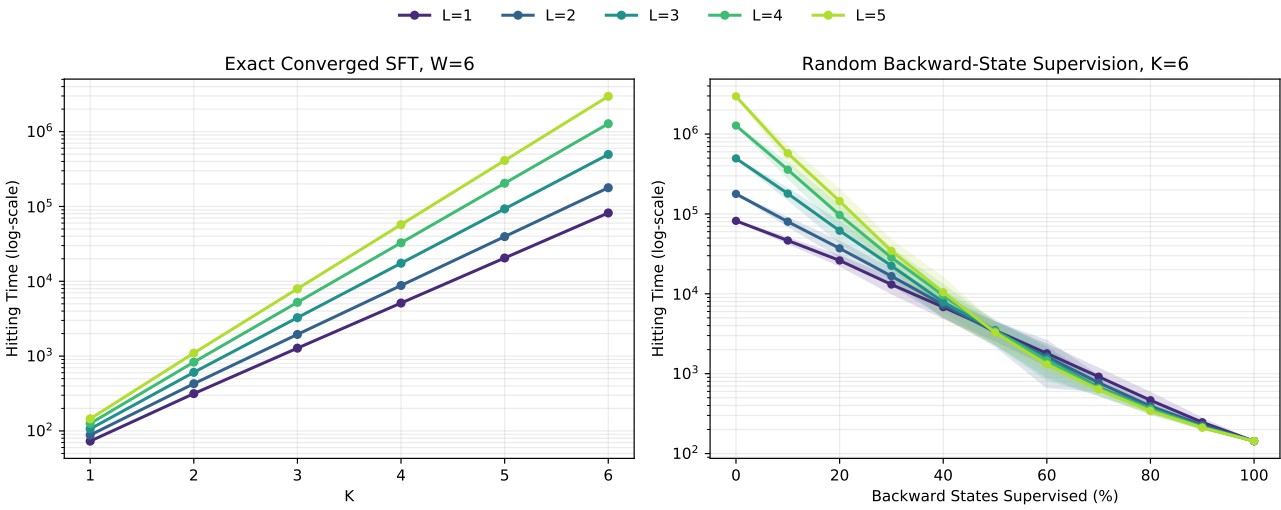

*Figure 5.* For our SFT experiments, we fix $W = 6$ branches, and plot the hitting times on a log scale. **Left:** For each $1 \leq L \leq 5$, we plot a curve of the (log-) hitting times with respect to the branch length $K$. **Right:** Here, we additionally fix $K = 6$. For $1 \leq L \leq 5$, we observe the empirical hitting times become upon additional supervision of backtracking data. That is, we randomly sample a $p$ fraction of backwards states $L_j^-$ and $R_j^-$ over all $1 \leq j \leq K$, give it supervision through SFT data, and plot the average hitting time under this policy where the supervised backwards states learn to continue backwards. For our experiments, we take average over 100 random supervisions for each $p = 0, 0.1, \ldots, 0.9, 1.0$.

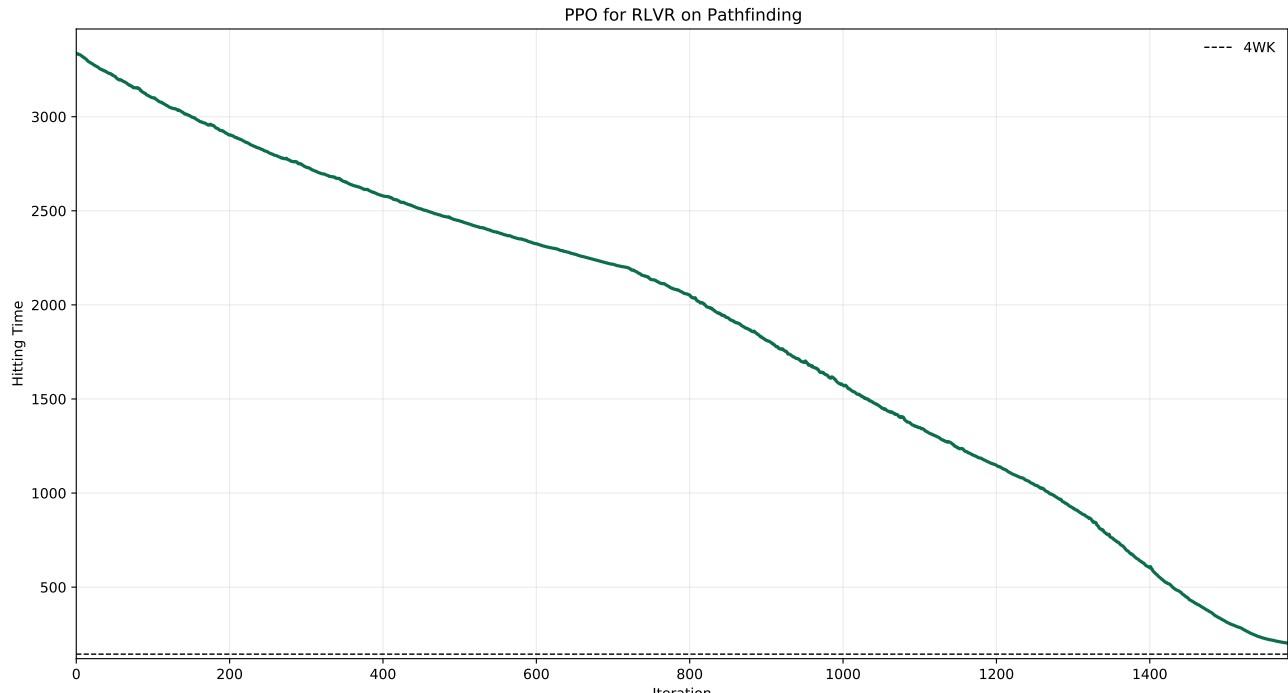

*Figure 6.* We run PPO on rollout samples for RLVR. Notably, for a given iteration, we use a rollout horizon of two times the current policy's hitting time of uniform targets, and a very small length penalty of $\beta = 3 \times 10^{-4}$. Our choice ensures we can observe the synergy between the horizon and the length penalty in a more realistic setting. Convergence to the hitting time of $4WK$ (the learned backtracking policy) is shown above.

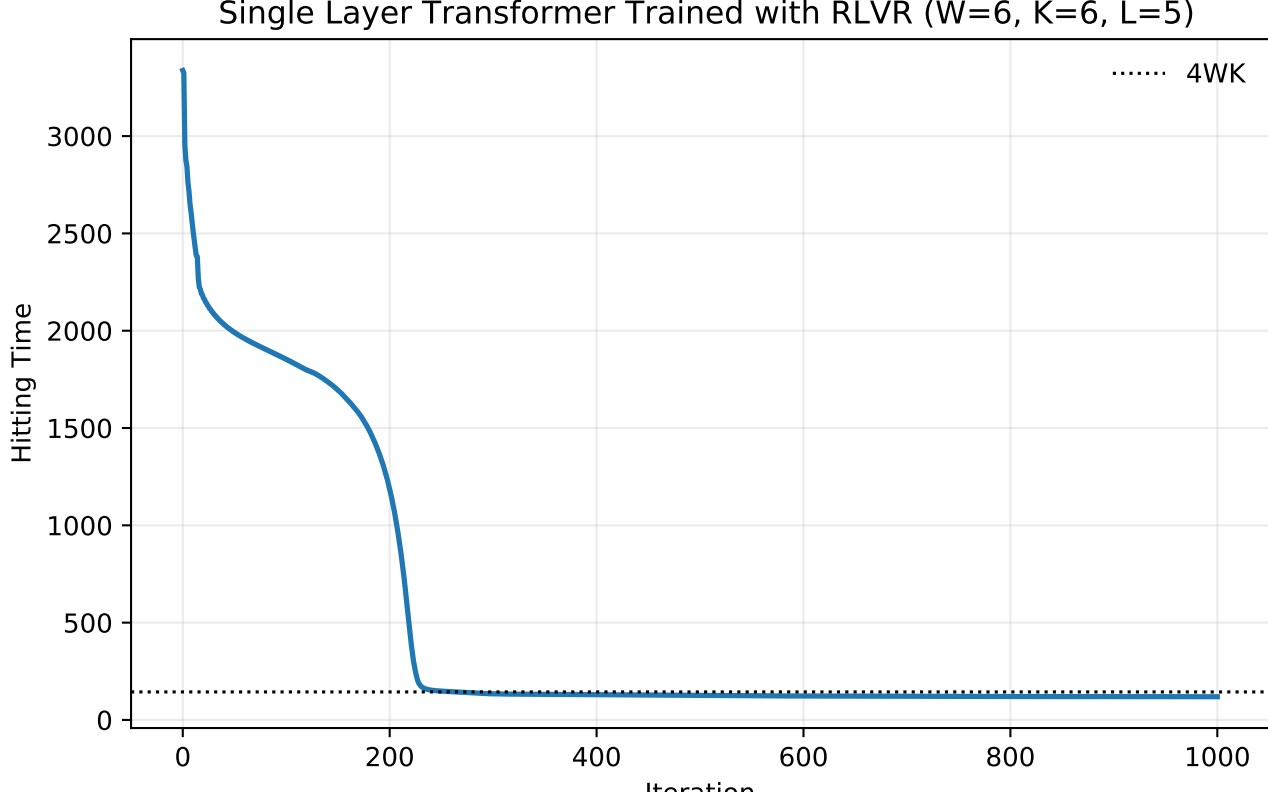

*Figure 7.* We simulate the training of a single-layer transformer on our pathfinding task. We initialize with the bigram policy as our pretrained policy as per our theory (e.g., by setting the attention matrix equal to identity), and run RLVR via the policy gradient. As predicted by our theoretical analysis, the policy is able to converge to the theoretical optimal of $4WK$ (in fact, it does slightly better than that since a transformer has more expressivity than a bigram model and hence may not have an exact bigram policy).

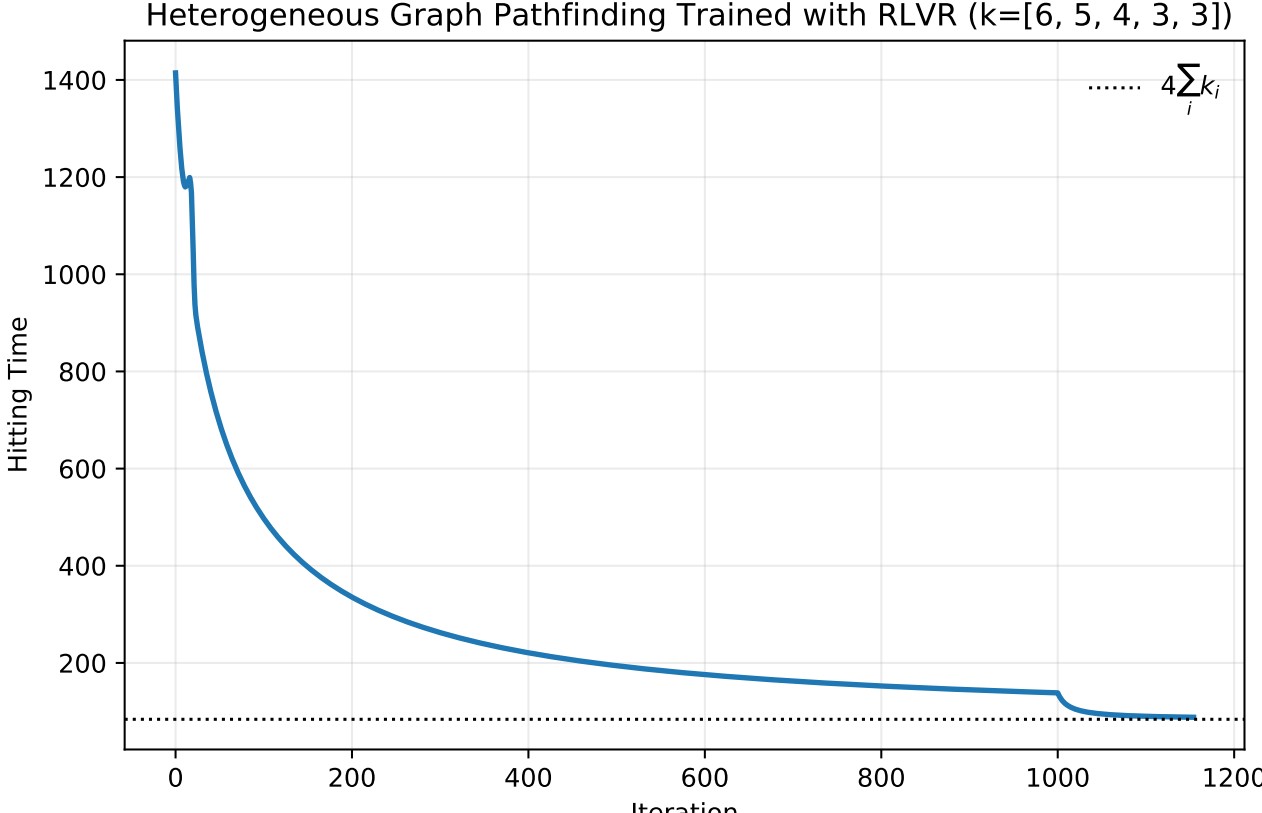

*Figure 8.* We go beyond the symmetric setting of our paper and analyze the convergence dynamics on a non-symmetric pathfinding task, where different branches are allowed to have different lengths, and different diamonds are allowed to have differing amounts of multiedges. Here, the backtracking target policy will have hitting time $4 \sum_i k_i$, instead of $4WK$. Indeed, RLVR on the policy gradient converges to this policy.

