# OpenReview forum: "Provable Benefits of RLVR over SFT for Reasoning Models: Learning to Backtrack Efficiently"
_ICML.cc/2026/Conference — ICML 2026 regular_

### Official Review · Reviewer_AfEk · 2026-03-11

**Soundness:** 3
**Presentation:** 3
**Significance:** 2
**Originality:** 3
**Overall Recommendation:** 4
**Confidence:** 3

**Summary:**

This paper analyzes, in a stylized but mathematically explicit graph pathfinding model of chain-of-thought, why reinforcement learning with verifiable rewards (RLVR) can endow a pretrained model with efficient backtracking abilities at test time, while supervised fine-tuning (SFT) on golden shortest-path traces does not. Using a linear softmax bigram policy with signed gradient flow, the authors prove that SFT cannot improve backward transitions because such states never appear in the data, whereas RLVR’s on-policy rollouts provide gradients that drive both forward and backward transitions toward efficient traversal and retreat, yielding a provable inference-time compute separation: $O(WK)$ for RLVR vs $O(W L^K)$ for SFT in a multi-branch, multi-diamond graph. They further show that distilling RLVR reasoning traces via SFT can transfer the efficient backtracking behavior.

**Compliance With Llm Reviewing Policy:**

Affirmed.

**Final Justification:**

The rebuttal addresses my concerns by incorporating experiments with asymmetric graph analysis. These additions strengthen the empirical evidence compared to the original paper.

Therefore, I have updated my assessment accordingly.

**Key Questions For Authors:**

### Questions

1. How sensitive are your RLVR dynamics and the O(WK) result to the length penalty β and to finite rollout horizons? Can you provide experiments varying β and horizon length?

2. Can you show empirical SFT hitting times versus K and L to corroborate the exponential scaling claim, and include at least one setting where SFT is augmented with a small amount of backtracking data?

**Limitations:**

Yes

**Strengths And Weaknesses:**

### Strengths

1. The work establishes a provable inference-time compute separation between RLVR and SFT under explicit assumptions.

2. The theory proposed addresses a timely and practically important question: why RL-style post-training improves reasoning and search compared to SFT on gold traces. Adding another explanation to the question is undoubtedly beneficial to the community. The result supports a widely observed empirical phenomenon and provides guidance for data/algorithm design (e.g., value of backtracking traces and distillation).


### Weaknesses

1. The model is highly stylized: a bigram softmax with signed gradient flow, exact pretraining to the world model, and a very symmetric multigraph. It is unclear how robust the separation is under finite-sample SGD, function approximation mismatch, or less symmetric graphs.
2. No empirical SFT baseline curve demonstrating the purported exponential blow-up; only RLVR dynamics are plotted. Besides, no sensitivity to length penalty $\beta$, rollout horizon, or stochasticity.

---

> ### Author Rebuttal · Authors · 2026-03-31
>
> We thank the reviewer for their insightful questions. Below, we address the concerns and provide additional experiments that we will also incorporate into the revision. Figures for responses to all reviewers are located [here](https://github.com/ICML2026-Submission3713/Submission3713/blob/main/rebuttal_plots.pdf).
> \
> \
> **Q1: How sensitive are your RLVR dynamics and the $O(WK)$ result to the length penalty $\beta$ and to finite rollout horizons? Can you provide experiments varying $\beta$ and horizon length?**
>
> Our theory analyzes the case where the either the horizon $H$ or the length penalty $\beta$ is unbounded, and hence reward advantages are governed by the rollout lengths due to the length penalty. When this is not the case, we empirically demonstrate in Figure 2 that when we lower the length penalty $\beta$ to being very small, a reasonably chosen horizon length allows for relatively stable optimization towards the desired backtracking policy.
>
> One caveat we would like to mention is that experiments on fixed horizon length $H$ throughout training will generally not be the most efficient. To see why, note that as training progresses, the rollout length will naturally decrease (due to learning of more efficient backtracking behavior). If the horizon is too short, it will be very hard to find rollouts that both hit the target and encounter sufficient backtracking data to learn. On the other hand, if the horizon is too long yet the length penalty $\beta$ is too small, it will be likely that the rollout hits the target but the policy does not get much gradient update. To marry these observations, Figure 2 is the result of using rollout horizon being twice the expected hitting time at any given iteration.
>
>
> **Q2: Can you show empirical SFT hitting times versus K and L to corroborate the exponential scaling claim, and include at least one setting where SFT is augmented with a small amount of backtracking data?**
>
> We refer the reviewer to Figure 1, which contain the SFT trained model's expected hitting times. Additionally, we also include simulations to see how hitting times change upon addition of varying amounts of backtracking data for SFT. In practical settings, it nonetheless remains an open question to understand what characteristics of backtracking traces help the most in terms of model training. For instance, [1] analyzes this question from an empirical perspective, in order to understand how much backtracking data is exactly necessary.
>
>
> **Q3: The model is highly stylized: a bigram softmax with signed gradient flow, exact pretraining to the world model, and a very symmetric multigraph. It is unclear how robust the separation is under finite-sample SGD, function approximation mismatch, or less symmetric graphs.**
>
>
> The goal of our symmetric setup is twofold: 1) it isolates the effect of learning a clean backtracking policy via RLVR, and 2) it already induces nontrivial optimization dynamics to analyze. However, for general distributions of training targets, as well as heterogeneous $K,L$, the learned probabilities of entry for each branch should slightly differ. Nevertheless, the behavior of "forward states continue forward; backward states continue backwards" should still be learned, as otherwise it would increase the hitting time.
>
> To verify this intuition, we run a new synthetic experiment (see Figure 4). Here, we consider a graph setup with heterogeneous $K,L$ across branches, and we see the hitting time still converges to the expected hitting time of $4\sum_i k_i$ (in our paper, all $k_i=K$) as suggested by this policy. We additionally deviate from the world model symmetry; besides running for a heterogeneous branch world model, we also add some light noise to the initial pretrained probabilities and observe that the convergence behavior occurs as our theory suggests. Experiments are run with PPO and on-policy rollouts.
> \
> \
> \
> We humbly ask the reviewer to consider raising their score if their concerns have been satisfactorily addressed.
> \
> \
> References:
>
> [1] Cai et al. (2025), How Much Backtracking is Enough? Exploring the Interplay of SFT and RL in Enhancing LLM Reasoning

---

> > ### Author Rebuttal · Reviewer_AfEk · 2026-04-08
> >
> > The rebuttal addresses my concerns by incorporating experiments with asymmetric graph analysis. These additions strengthen the empirical evidence compared to the original paper.
> >
> > Therefore, I have updated my assessment accordingly.

---

> > > ### Author Response · Authors · 2026-04-08
> > >
> > > We thank the reviewer for the positive feedback in their rebuttal acknowledgement. We agree that these broader empirical setups will enhance our work, and we look forward to incorporating our discussions and new experimental results into the revised manuscript.

---

### Official Review · Reviewer_k2UC · 2026-03-13

**Soundness:** 2
**Presentation:** 2
**Significance:** 2
**Originality:** 3
**Overall Recommendation:** 3
**Confidence:** 4

**Summary:**

This paper focuses on the core problem of backtracking ability in LLMs' CoT, and conducts a theoretical comparison between two mainstream post-training methods: SFT and RLVR. To quantify reasoning and backtracking behaviors, the authors model CoT reasoning as a pathfinding problem on a structured multigraph and formalize the LLM as a single-layer softmax bigram model with state transitions defined by logit parameters. Finally, the paper identifies the lack of backtracking data in golden paths as the root cause of SFT’s inefficiency, and proves that RLVR’s on-policy exploration provides critical gradient signals for learning backtracking.

**Compliance With Llm Reviewing Policy:**

Affirmed.

**Final Justification:**

I would like to reiterate one remaining concern: I still feel the scope of this paper is somewhat narrow, and as a result it may not provide substantial direct insight into practical mechanisms or design choices in real-world SFT/RL training setups. That said, given that theoretical work in the LLM reasoning community is still relatively slow to develop, the paper’s theoretical framing and provable results do add meaningful value.

**Key Questions For Authors:**

> Theoretical

- Can the exponential inference-time separation conclusion (Θ(LK)) derived from the bigram model be generalized to transformer models (the backbone of real LLMs)?
- For asymmetric reasoning structures with variable K/L across branches, non-uniform multiedge weights, do the convergence characteristics of RLVR and the backtracking ability still hold?
- How do verifier errors/bias affect RLVR’s ability to learn backtracking?

> Practical

- For practical RL optimizers (e.g., PPO, GRPO) instead of the theoretical signed gradient flow, how does the training stability of RLVR affect the learning of backtracking ability?
- What is the compute trade-off between RLVR’s training cost (on-policy rollouts) and inference efficiency gain? For different reasoning task scales (small/medium/large K/L), is there a critical point where the inference efficiency gain of RLVR offsets its higher training compute cost?

> Others
- For SFT models, can we augment golden path data with artificial backtracking traces to achieve the same backtracking ability as RLVR? What characteristics of backtracking traces are the most critical for SFT training?
- How does the inference-time search framework (e.g., Tree-of-Thoughts, Graph-of-Thoughts) interact with the backtracking ability learned via post-training?

**Limitations:**

Yes

**Strengths And Weaknesses:**

> Strengths

- Prior works on SFT and RLVR for LLM reasoning are mostly empirical; this paper is a theoretical analysis to provide a dynamical and provable comparison.
- The overall framework is easy to read, and the assumptions and definitions are clear to me.
- The paper quantifies the efficiency gap of  RLVR and SFT from the perspective of backtracking.

> Weaknesses

- The paper models the LLM as a single-layer softmax bigram model with only edge/ direction states, which is far from modern LLMs.
- All theoretical conclusions are verified only on a synthetic, structured multigraph pathfinding task, with no experiments on real-world reasoning tasks.
- The paper makes several idealized assumptions for RLVR analysis: (1) the verifier is perfect (stops rollouts exactly at the target node, no verification error); (2) the length penalty β is optimally chosen and fixed; (3) RLVR is optimized via signed gradient flow (a theoretical optimizer), rather than practical RL. (4) the policy is independent of the prompted target.
- The paper ignores the cost of RLVR’s on-policy exploration, on-policy RL requires continuous re-sampling of rollouts with the latest policy, which incurs much higher training compute cost than off-policy SFT.
- All theoretical analyses are based on a strictly symmetric multigraph (all branches have the same K/L, all multiedges have the same transition probability). In real reasoning tasks, reasoning paths are often asymmetric (e.g., some problem-solving steps have more choices, some branches are shorter/longer), and the paper does not explore whether the topological symmetry is a necessary condition for the conclusions.

---

> ### Author Rebuttal · Authors · 2026-03-31
>
> We thank the reviewer for their detailed and insightful questions, and we hope our responses and experiments sufficiently address the reviewer's concerns. Figures for responses to all reviewers are located [here](https://github.com/ICML2026-Submission3713/Submission3713/blob/main/rebuttal_plots.pdf).
> \
> \
> **Q1: Can the exponential inference-time separation conclusion derived from the bigram model be generalized to transformer models?**
>
> We expect the same results will hold for transformers, as suggested by fact that transformers can learn causal structure per [1]. This is because our separation is not inherently tied to the bigram parameterization, but rather any higher-capacity model that realizes this local transition policy. Formally, suppose that our pretrained transformer model learns the bigram world model of local edge transitions in our paper. Then, the RLVR fine-tuned model will similarly learn the policy in our paper, whereas the SFT model, having never encountered the backwards states, will fail to generate meaningful backtracking behavior. We have included a single-layer transformer experiment for RLVR; see Figure 3.
>
>
> **Q2: For asymmetric reasoning structures with variable K/L across branches, non-uniform multiedge weights, do the convergence characteristics of RLVR and the backtracking ability still hold?**
>
> We refer the reviewer to our response to Q3 of reviewer AfEk; see Figure 4 for the relevant experiment.
>
>
> **Q3: How do verifier errors/bias affect RLVR’s ability to learn backtracking?**
>
> This question goes beyond the scope of our paper; for instance, learning from noisy feedback is an active area of research [2, 3]. There is also no one-size-fits-all definition of verifier errors. Qualitatively, we would expect small unbiased verifier errors to be relatively benign, while systemic biases on certain branches of the task may interfere with learning of the advantage function. We will add this as a point of discussion in a revised version of our paper.
>
>
> **Q4: For practical RL optimizers instead of the theoretical signed gradient flow, how does the training stability of RLVR affect the learning of backtracking ability?**
>
> We refer the reviewer to our response to Q1 brought up by reviewer AfEk; see Figure 2 for the relevant experiment.
>
>
>
> **Q5: What is the compute trade-off between RLVR’s training cost and inference efficiency gain? For different reasoning task scales, is there a critical point where the inference efficiency gain of RLVR offsets its higher training compute cost?**
>
> In our theoretical analysis, there is no clear comparison of training vs. inference cost; for instance, our paper analyzes the regime where the dynamics follow a continuous-time flow, which is not directly comparable with the our metric of measuring inference-time compute as the number of tokens (e.g., hitting time). Even in practice, post-training happens at much longer timescales compared to inference, so in general any nontrivial amount of improvement in test-time performance is desirable.
>
>
> **Q6: For SFT models, can we augment golden path data with artificial backtracking traces to achieve the same backtracking ability as RLVR? What characteristics of backtracking traces are the most critical for SFT training?**
>
> We refer the reviewer to our response to Q2 of reviewer AfEk; see Figure 1 for the relevant experiment.
>
>
> **Q7: How does the inference-time search framework (e.g., Tree-of-Thoughts, Graph-of-Thoughts) interact with the backtracking ability learned via post-training?**
>
> We currently assume no inference-time search for the models; we are purely comparing the ability of a single autoregressive generation. Suppose now that both models are given access to an inference-time search framework, which backtracks a step upon reaching an already visited edge state. Because RLVR already fully learns the backtracking ability, the additional benefits of this search framework will be negligible, so the expected hitting time is still $\Theta(WK)$. On the other hand, while the SFT model will have significant gains in terms of inference time efficiency (i.e. the hitting time is no longer exponential), there remains a big separation between it and the RLVR model. In particular, the SFT model with search will have expected hitting time $\Theta(WKL)$, which is still a factor of $L$ more. This is due to the $L$ multiedges, for which the SFT backward edge states cannot distinguish with the actual backwards continuation. We will formalize this as an additional theorem in our paper, and we thank the reviewer for pointing out this practical connection.
> \
> \
> \
> We humbly ask the reviewer to consider raising their score if their concerns have been satisfactorily addressed.
> \
> \
> References:
>
> [1] Nichani et al. (2024), How Transformers Learn Causal Structure with Gradient Descent
>
> [2] Xu et al. (2025), Provably Learning from Language Feedback
>
> [3] Song et al. (2026), Expanding the Capabilities of Reinforcement Learning via Text Feedback

---

> > ### Author Rebuttal · Reviewer_k2UC · 2026-04-04
> >
> > Thank you for your detailed rebuttal.
> >
> > - The rebuttal partially addresses my concerns by adding experiments with PPO/on-policy rollouts and extending to asymmetric graphs. These additions make the empirical phenomenon appear more robust than in the original submission.
> >
> > - My main concern is not merely that the analysis is simplified, but that the central separation relies on several structural assumptions: a local bigram-like policy class, a graph-search abstraction of reasoning, a perfect verifier, and the absence of meaningful backward-state supervision in SFT data. These assumptions are not cosmetic, It define the very mechanism by which RLVR outperforms SFT, and therefore limit how strongly the result can be interpreted for modern transformer-based LLMs.
> >
> > - For example, a core limitation is the strong bigram assumption. The paper models reasoning as a local transition policy where the current state determines the next step, which is much closer to a single-layer softmax bigram model than to a modern transformer. In practice, backtracking in LLMs often depends on long-range context, prior error patterns, and task semantics rather than only the local edge state. Thus, **the claim that SFT cannot backtrack because backward states are never observed may be specific to this restricted model class, rather than a general conclusion about LLMs.**
> >
> > - Another core assumption is the availability of a perfect verifier, which effectively provides RLVR with ideal supervision through the reward signal. Yet the ability to learn backtracking hinges on whether the reward/advantage accurately identifies **when backtracking is appropriate**. In more realistic settings, verifier bias, or local verification errors could distort this signal, potentially weakening or even overturning the claimed advantage of RLVR.

---

> > > ### Author Response · Authors · 2026-04-04
> > >
> > > Thank you for continuing to engage in the discussion. We respectfully argue that these assumptions are necessary to obtain a fine-grained dynamical analysis. While it would of course be ideal to relax all assumptions, the theoretical understanding of post-training is still a nascent field, and most cutting-edge papers either use similar assumptions or have to take a black-box approach (e.g. RL-theoretic approaches). We address each in turn, giving ample such examples.
> > >
> > > - **Bigram policy class:** While the bigram or Markov model is restrictive, it has been used multiple times in state-of-the-art dynamical analysis papers on LLM reasoning [1,2,3,4] and has also been used to guide practical CoT generation [5]. It is exceedingly difficult to analyze the dynamics of training on multi-step traces without such simplifying assumptions, and the bigram model is already capable of giving nontrivial insights into post-training as evidenced by these prior works. In contrast, other theoretical works which allow for general autoregressive models cannot study the inner dynamics of the models themselves, and usually only obtain statistical guarantees (classical sample complexity or regret bounds).
> > >
> > > - **Graph-search abstraction:** Graph search, connectivity, shortest-path, etc. have long been used as a sandbox to explore the reasoning capabilities of LLMs both theoretically and empirically; [6,7,8,9] just to name a few. These include algorithmic complexity analyses for general graphs, as well as in-depth case studies of specific graphs such as the famous path-star graph [7] and bridge graph [8]. Our work can be seen as extending the latter body of work with a rigorous dynamical analysis of our fork multigraph.
> > >
> > > - **Absence of backward supervision in SFT:** Rather than an unrealistic assumption on real-world SFT data, this condition should be interpreted as **establishing the importance of backward supervision even in SFT**. First note that if we allowed arbitrary paths in SFT data, there is nothing stopping the SFT expert from showing the learner the search traces collected from the RL learner (incl. backtracking) and mimicking RL, so it is impossible to establish a separation; hence one must impose some sort of handicap on SFT to get any meaningful conclusion. Conversely, by proving that SFT without backtracking is suboptimal, we are establishing the importance of backtracking in both RLVR and SFT (not simply "RLVR is always better"). This has immediate practical takeaways; SFT expert data is often curated without backtracking, e.g., when mathematicians create solutions to problems, they very rarely include wrong or exploratory paths in the solution; our results show that reasoning capability can benefit from explicitly including such traces. We will make this point clearer in the text.
> > >
> > > - **Perfect verifier:** We agree that this is an interesting additional direction for future work, but believe that this is not a critical limitation of the current scope of our work; this is a genuinely difficult question and a whole research topic on its own. Indeed, papers allowing for noisy rewards such as [11] typically assume strong conditions such as per-step independence which makes the effect of error benign (not accumulate over long horizons). We are aware of only one recent paper [10] which analyzes the impact of imperfect verifiers on multi-step autoregressive processes in depth. Moreover, since we only consider outcome reward, the model is not directly given information on when and where to backtrack as is the case for the backtrack oracle in [6], which we agree can be seen as too strong of an assumption.
> > >
> > > We again thank the reviewer for the thoughful discussion, and will add these points in the manuscript to better situate our work.
> > >
> > > [1] Kim et al., 2025. Metastable Dynamics of Chain-of-Thought Reasoning: Provable Benefits of Search, RL and Distillation.
> > >
> > > [2] Wang et al., 2026. When does Chain-of-Thought Help: A Markovian Perspective.
> > >
> > > [3] Bu et al., 2026. Post-Training as Reweighting: A Stochastic View of Reasoning Trajectories in Language Models.
> > >
> > > [4] Nichani et al., 2024. How Transformers Learn Causal Structure with Gradient Descent.
> > >
> > > [5] Yang et al., 2025. Markov Chain of Thought for Efficient Mathematical Reasoning.
> > >
> > > [6] Shalev-Shwartz et al., 2025. From Reasoning to Super-Intelligence: A Search-Theoretic Perspective.
> > >
> > > [7] Frydenlund, 2024. The Mystery of the Pathological Path-star Task for Language Models.
> > >
> > > [8] Mirtaheri et al., 2025. Let Me Think! A Long Chain-of-Thought Can Be Worth Exponentially Many Short Ones.
> > >
> > > [9] Xiong et al., 2025. Mapping the Minds of LLMs: A Graph-Based Analysis of Reasoning LLMs.
> > >
> > > [10] Rohatgi et al., 2025. Taming Imperfect Process Verifiers: A Sampling Perspective on Backtracking.
> > >
> > > [11] Setlur et al., 2025. Scaling Test-Time Compute Without Verification or RL is Suboptimal.

---

### Official Review · Reviewer_JwDT · 2026-03-13

**Soundness:** 4
**Presentation:** 2
**Significance:** 4
**Originality:** 4
**Overall Recommendation:** 5
**Confidence:** 3

**Summary:**

This paper theoretically analyzes the difference between the reasoning abilities induced by Reinforcement Learning (GRPO) and Supervised Fine-tuning on LLMs, especially the backtracking abilities that exist in RL post-trained LLMs but not by SFT. The paper utilizes a graph-based pathfinding problem formulated as a reasoning problem to compare the two fine-tuning paradigms and show how RLVL trained model can learn to backtrack from dead ends only using outcome rewards while SFT can not.

**Compliance With Llm Reviewing Policy:**

Affirmed.

**Final Justification:**

Given the rebuttal discussion, I will like to maintain my positive assessment.

**Key Questions For Authors:**

Please see the weaknesses section.

**Limitations:**

Yes

**Strengths And Weaknesses:**

**Strengths:**

1. This is an interesting paper that can significantly benefit and aid further discussions in the community for gaining a better understanding of these fine-tuning paradigms. The paper follows a very principled approach to compare and theoretically analyze the two algorithms and the differences observed in the trained LLM.

2. Using a pathfinding reasoning problem is a good choice to cleanly analyze the effect of the two training regimes especially with respect to drawing conclusions from the learnt state transitions (to understand backtracking).

**Weaknesses:**

1. While this work does not represent backtracking via phrases such as 'Let us think again' or 'Reflect' etc. as has been anthropomorphized for backtracking behaviors in the literature, it still mentions that in the Introduction section. It would be helpful to clarify if the paper takes a stance on considering the explicit token usage as backtracking as well under the same paradigms.

2. It is also not clear how the study design and results can be easily understood to translate to text-based reasoning tasks which LLMs are fine-tuned for more commonly. While it is understood why the same analysis may not be feasible for such tasks, it will be important to have some discussion on the same.

3. How would this result translate to process-based reward models?

---

> ### Author Rebuttal · Authors · 2026-03-31
>
> We thank the reviewer for their positive assessment of our work. Below, we address the aforementioned comments and questions.  Figures for responses to all reviewers are located [here](https://github.com/ICML2026-Submission3713/Submission3713/blob/main/rebuttal_plots.pdf).
> \
> \
> **Q1: While this work does not represent backtracking via phrases such as 'Let us think again' or 'Reflect' etc. as has been anthropomorphized for backtracking behaviors in the literature, it still mentions that in the Introduction section. It would be helpful to clarify if the paper takes a stance on considering the explicit token usage as backtracking as well under the same paradigms.**
>
> Our notion of backtracking is operational rather than lexical. As backtracking phrases do not explicitly return the generation to an earlier decision point, our work is more closely aligned with methods which explicitly introduce rollback operations in reasoning; for instance see the self-backtracking mechanism proposed in [1]. That said, we do not claim that `in-text' backtracking can or cannot replace such external mechanisms. Our main message is that some amount of backtracking data is necessary during training (whether through supervision in SFT data, or from encountering it during RLVR); understanding the extent that the above phrases help in acquiring this ability is an open question for empirical study. For an example of how the backtracking data can affect hitting times in the context of our setup, we refer the reviewer to Figure 1. We will include a more detailed discussion on this connection in a revision of our paper.
>
>
> **Q2: It is also not clear how the study design and results can be easily understood to translate to text-based reasoning tasks which LLMs are fine-tuned for more commonly. While it is understood why the same analysis may not be feasible for such tasks, it will be important to have some discussion on the same.**
>
> We definitely agree with this sentiment; unfortunately real-world text-based reasoning cannot be modeled cleanly as in our theoretical pathfinding setting. Nevertheless, we believe there is still merit in studying the star graph setting. As an example, mathematical reasoning can be seen as a close abstraction to a pathfinding task, with edge transitions representing steps taken to compose hypotheses or deduce new lemmas. We will make this connection clearer in the revision. We hope that analyzing the (already nontrivial) dynamics on our toy setting will shed insight in how the practical dynamics occur.
>
>
>
> **Q3: How would this result translate to process-based reward models?**
>
> Our setup intentionally studies the sparse (outcome-based) reward case, since we can already show it is sufficient for the emergence of backtracking ability on our pathfinding task. We would imagine that a process reward model would provide denser intermediate signal, and can accelerate convergence signal to this learned backtracking behavior for RLVR. Nevertheless, we show in our setting that it is not always necessary (and in practice, PRMs are more expensive to train).
> \
> \
> \
> References:
>
> [1] Yang et al. (2025), Step Back to Leap Forward: Self-Backtracking for Boosting Reasoning of Language Models.

---

> > ### Author Rebuttal · Reviewer_JwDT · 2026-04-03
> >
> > Thanks for addressing the concerns and questions, I will keep my positive assessment.

---

> > > ### Author Response · Authors · 2026-04-04
> > >
> > > We thank the reviewer again for their rebuttal acknowledgement. On that note, we wanted to apologize for any inconveniences caused by our broken initial link to the additional experiments. If the reviewer would like to see additional experimental results at any point during the discussion period, please let us know, and we will add/update the figures in https://anonymous.4open.science/r/Submission-3713-ADA5/rebuttal_plots.pdf accordingly.

---

### Decision · Program_Chairs · 2026-04-30

**Decision:**

Accept (regular)

**Comment:**

The work shows a new theoretical result, comparing SFT and RLVR for path-finding on a graph search problem. The authors show that while SFT fails to learn backtracking in this setting, RLVR succeeds in learning to backtrack.

The reviewers overall appreciated the theoretical contribution, mentioning the principled approach taken by the paper for comparing two popular post-training methods - SFT and RLVR. Some reviewers found the graph search problem to be a clean and interesting setting for analyzing how reasoning is learned by different methods. One criticism that was shared between the reviewers is the limited setting in the paper, and in particular the restricted policy class that was used for the theoretical analysis. Reviewers raised concerns regarding the generalizability of the theoretical results to more practical settings and classes. However, after further discussion, reviewers agreed that the restricted setting makes the theory cleaner, and the theoretical results in the current setting are valuable.

In light of this, I recommend that this paper is accepted, but encourage the authors to address the concerns raised by the reviewers in the final version of the paper. Specifically, the authors should incorporate a clear discussion on the limitation that arises from the restricted setting, and acknowledge that their results may or may not hold for more general settings.